



# Constraining regional glacier reconstructions using past ice thickness of deglaciating areas – a case study in the European Alps

Christian Sommer[1], Johannes J. Fürst[1], Matthias Huss[2,3,4] & Matthias H. Braun[1]

[1]Institut für Geographie, Friedrich-Alexander-Universität Erlangen-Nürnberg, 91058 Erlangen, Germany
[2] Swiss Federal Institute for Forest, Snow and Landscape Research (WSL), Birmensdorf, Switzerland
[3] Laboratory of Hydraulics, Hydrology and Glaciology (VAW), ETH Zürich, Zürich, Switzerland
[4] Department of Geosciences, University of Fribourg, Fribourg, Switzerland

*Correspondence to*: Christian Sommer (chris.sommer@fau.de)

**Abstract.** In order to assess future glacier evolution and melt-water runoff, accurate knowledge on the volume and the ice thickness distribution of glaciers is crucial. However, in-situ observations of glacier thickness are sparse in many regions worldwide due to challenging field surveys. This lack of in-situ measurements can be partially overcome by remote sensing information. Multi-temporal and contemporaneous data on glacier extent and surface elevation provide past information on ice thickness for retreating glaciers in the newly deglacierized regions. Yet, these observations are concentrated near the glacier snouts, which is disadvantageous because it is known to introduce biases in ice thickness reconstruction approaches. Here, we show a strategy to overcome this generic limitation of so-called "retreat" thickness observations by applying an empirical relationship between the ice viscosity at locations with in-situ observations and observations from DEM-differencing at the glacier margins. Various datasets from the European Alps are combined to model the ice thickness distribution of Alpine glaciers for two timesteps (1970 & 2003) based on observed thickness in regions uncovered from ice during the study period. Our results show that the average ice thickness would be substantially underestimated (~40%) when relying solely on thickness observations from previously glacierized areas. Thus, a transferable topography-based viscosity scaling is developed to correct the modeled ice thickness distribution. It is shown that the presented approach is able to reproduce region-wide glacier volumes, while larger uncertainties remain at a local scale, and thus might represent a powerful tool for application in regions with sparse observations. Additionally, we derive a volume of 125.4±24.7 km³ in the 1970s for glaciers in the Swiss and Austrian Alps.

## 1 Introduction

Glaciers are retreating in most mountain regions of the world due to climate warming. Recent measurements of global glacier change show that around 20% of the observed sea-level rise during the 21st century can be attributed to mass loss of mountain glaciers (Hugonnet et al., 2021). Also, diminishing glacier volumes affect seasonal water runoff and the availability of fresh water (Huss and Hock, 2018; Rodell et al., 2018), particularly in arid and semiarid regions. On a local scale, glacier retreat



induces natural hazards related to periglacial and glacial environments (Stoffel and Huggel, 2012), such as rockfalls or flooding, but could also offer new hydrological storage and sustainable energy potentials (Ehrbar et al., 2018; Farinotti et al., 2019b). Therefore, knowledge of the distribution of glacier ice volume and thickness is crucial to predict future glacier retreat

and deglacierization as well as the subsequent consequences on freshwater supply, hazards (GLOFs) and sea-level (Marzeion et al., 2012). While increasingly detailed glacier area inventories are becoming available (Pfeffer et al., 2014), there are still no direct measurements of ice thickness for the majority of the glaciers (Welty et al., 2020). Furthermore, modelling approaches of the glacier-wide thickness distribution and volume are also required for glaciers with direct observations of ice thickness as in-situ measurements typically only cover a fraction of the glacierized area (Farinotti et al., 2021).

To efficiently derive the thickness of glaciers on a regional scale, the Ice Thickness Model Intercomparison eXperiment (ITMIX) aimed at the comparison of different thickness reconstruction approaches, solely based on properties of the glacier surface (Farinotti et al., 2017). Participating models were diverse and relied either on mass conservation, on simplifications of the force balance, on the perfect plasticity assumption or a combination of these. Historically, many of the approaches did not aim at reproducing available thickness measurements. These primarily entered as loose calibration or validation observations.

During the second experimental phase (ITMIX2), the intercomparison was therefore extended to include ice thickness measurements and tested the capability of these approaches to assimilate thickness observations. While the inclusion of thickness observations did improve the average modeled ice thickness, the results suggest that an uneven distribution of observations across the glacier domain can cause a systematic bias in ice thickness. Particularly, an underestimation of average glacier thickness was found for several models when relying preferentially on thickness observations of the lowest glacier

elevations. Contrasting, measurements of the thick glacier parts reduced the spread in estimated mean thickness between the ITMIX2 members (Farinotti et al., 2021). Similarly, a recent study, based on remote sensing glacier velocity measurements and an inversion approach of Stokes ice flow mechanics (Jouvet, 2022), showed that the availability of ice thickness observations, while less important for estimates of total glacier volumes, can greatly improve the modelled ice thickness distribution.

However, this poses a problem for thickness estimations of many glaciers as in-situ observations of ice thickness, such as direct measurements by drilling, seismic soundings or Ground Penetrating Radar (GPR), are usually associated with considerable logistical efforts and technical challenges. Ice thickness for a given previous glacier geometry is however readily available for the deglacierized region from remote sensing. Glacier inventories and digital elevation models (DEM) provide information on the extent and surface elevation. When comparing glacier outlines at different timesteps, once glacierized areas

can be identified which became ice-free between the acquisition dates of glacier inventories. Then, the original ice thickness is estimated by differencing the respective DEMs. With the growing number of available glacier outline and elevation datasets for different moments of time (Paul et al., 2020), these observations, termed "retreat thickness observations" henceforth, will increasingly become available. This information is a large asset for calibrating reconstruction approaches in many regions without in-situ thickness measurements. Even though some local inconsistencies related to changes in terrain elevation after

deglaciation due to erosion and sedimentation processes might be present, the uncertainty in surface elevation inferred by



remote sensing is typically much smaller than a direct measurement of ice thickness, e.g. by GPR, and complete information on former ice thickness in deglaciated areas is available. Therefore, retreat thickness observations have a considerable potential to improve ice thickness estimates for the entire glacier.

Here, we present an approach to reconstruct glacier-wide ice thickness distribution and volume from thickness observations based on repeated DEMs in deglacierized areas. To avoid a potential underestimation of the mean ice thickness, the model is calibrated with a slope- and elevation-based rescaling of the ice viscosity. The reconstruction is based on Alpine glaciers as in the European Alps glacier monitoring activities are more intense and denser than anywhere else in the world (Haeberli et al., 2007; WGMS, 2021). We use glacier inventories and DEMs to identify areas at the glacier margins which became ice-free since the 1970s and extract the prior ice thickness by differencing respective DEMs. Additionally, empirical viscosity scaling parameters are derived from a large number of available in-situ measurements of ice thickness, compiled from various sources. Then, the 1970s ice thickness distribution of Swiss and Austrian glaciers is reconstructed based on remote sensing data of the period ~1970-2019 and different subsets of thickness observations from field surveys and deglacierized areas. Finally, the approach is transferred to all alpine glaciers and the modeled ice thickness of the early 21$^{st}$ century is compared to previous reconstructions of Alpine glacier volumes.

## 2 Data & Methods

The ice thickness distribution of Alpine glaciers is calculated following the ice thickness reconstruction proposed by Fürst et al. (2017, 2018). The reconstruction is based on mass conservation, adapted from (Morlighem et al., 2011), and relies on the principle of the shallow ice approximation (SIA) (Hutter, 1983). Initially, the glacier-wide ice flux is derived from the difference of the surface mass balance (SMB) and the surface elevation change. Thereafter, the estimated ice flow is converted to ice thickness assuming the SIA. SIA based approaches are used by a number of recent regional to global glacier thickness reconstructions (Farinotti et al., 2019a; Maussion et al., 2019; Millan et al., 2022). The here applied reconstruction by Fürst et al. (2017) showed a close resemblance of locally observed ice thickness and robust thickness estimates during the ITMIX 2 intercomparison if thickness observations were provided (Farinotti et al., 2021).

### 2.1 Ice thickness reconstruction

The mass conservation of the ice flow is expressed as vertical integral following Eq. (1) (Cuffey and Paterson, 2010):

(1)  $\frac{\delta H}{\delta t} + \nabla \cdot (\mathrm{u}H) = \mathfrak{b}_S + \mathfrak{b}_b$





where $\nabla$ is the two-dimensional divergence operator, $H$ is the ice thickness, u are the vertically averaged, horizontal velocity

components and $\frac{\delta H}{\delta t}$ are the glacier surface height changes. $\mathfrak{b}_S$ and $\mathfrak{b}_b$ are the surface and basal mass balance. The product of u and $H$ is equal to the ice flux $F$.

Eventually, the glacier-wide flux $F$ is translated into local ice thickness according to Eq. (2) assuming the SIA (Hutter, 1983).

$$(2) \quad F^* = \frac{2}{n+2} B^{-n} (\rho g)^n ||\nabla h||^n \cdot H^{n+2}$$


The two-dimension flux field solution is determined over the entire drainage basin, as defined by the glacier compound outline. Here, $n$ is the flow law exponent, $\rho$ the density of ice (917 kgm$^{-3}$), $g$ the gravitational acceleration (9.18 ms$^{-2}$) and $\eta$ the ice viscosity.

**2.2 Viscosity scaling**

Motivated by spatial uncertainties when estimating ice thickness from lateral measurements (Farinotti et al., 2021), the mass conservation approach was extended by including dependencies of ice viscosity and surface slope (Carrivick et al., 2016) and elevation. Therefore, $\eta$ is calibrated according to slope and elevation dependant scaling factors (Eq. (3)):

$$(3) \quad \eta = \eta_0 \cdot \left( \xi_\eta^{slope} \cdot \xi_\eta^{elevation} \cdot \xi_\eta^{distance} \right)$$


$\eta_0$ is the initially estimated viscosity which is multiplied by correction factors for the surface slope $\xi_\eta^{slope}$ (section 3.2) and glacier elevation range $\xi_\eta^{elevation}$ (section 3.3). $\xi_\eta^{distance}$ is an additional scaling factor, based on the distance to the glacier margin (Eq. (4)):

$$(4) \quad \xi_\eta^{distance} = atan\left(\frac{d_{margin}}{4 \cdot H}\right) \times \frac{2}{\pi}$$

Where $d_{margin}$ is the distance to the glacier margin.

The ice viscosity $\eta_0$ is initially estimated at locations where ice thickness measurements are available. Then, $\eta_0$ is interpolated across the glacier domain. To better constrain $\eta_0$ at the domain margins and avoid extrapolation artefacts, the mean viscosity

from all measurements is prescribed around the glacier outline. For glaciers without any thickness measurements, the mean regionwide viscosity is used.



## 2.3 Uncertainty estimate

The uncertainty of the reconstructed ice thickness distribution and derived glacier volumes is estimated based on a formal error map. These error maps include contributions from the uncertainties of the input SMB (section 2.4.5) and $\frac{\delta H}{\delta t}$ (section 2.4.2)

information. Using those uncertainties, the flux error is estimated and converted into thickness uncertainty fields (Eq. (2)). At locations with ice thickness observations, the thickness error map is set to the respective measurement uncertainty. To avoid artifacts in the error maps, unrealistically high uncertainty values are replaced by the glacier-specific median ice thickness in cases where the local uncertainty is higher than the median ice thickness. A detailed description of the formal error estimation can be found in Fürst et al. (2017).

## 2.4 Input datasets

### 2.4.1 Ice surface elevation

Concerning the elevations of Alpine glaciers, past and present surface heights are derived from aerial photography, digitized topographic maps and space-borne SAR DEMs. For the Austrian Alps, aerial photographs of all glacierized areas were acquired with a mean picture scale of 1:30,000 between September and October 1969 (Patzelt, 1980). The original images were later

digitized and co-registered to derive the photogrammetric DHM69 elevation model (Lambrecht and Kuhn, 2007) which is used in this study. Historic glacier elevations of the Swiss Alps are available via the DHM25 dataset provided by Federal Office of Topography swisstopo (Anon, 2005). The DHM25 elevation model was created from the Swiss national topographic map (scale 1:25,000) by digitizing contour lines and spot heights which were then interpolated to a 25m resolution grid. We use the original DHM25lvl1 product, because glacier areas were updated with surface heights from winter 2000/01 in the

DHM25lvl2. Most map tiles covering the central Swiss Alps are from the 1980s (Anon, 2005) but the dates of glacier heights differ from the DHM25 specifications. Therefore, we refer to a detailed manual reconstruction (Fischer et al., 2015) of the specific reference years (1961-1991) of the DHM25 to derive the map date of each glacier. For the early 21$^{st}$ century, glacier surface topography is extracted from the 1 arcsec void-filled C-Band SAR DEM of the Shuttle Radar Topography Mission (Farr et al., 2007; Podest and Crow, 2013) which was acquired during February 2000. Recent glacier surface heights are

provided by the bistatic TanDEM-X satellite mission, operated by the German Aerospace Center (DLR) (Krieger et al., 2007; Zink et al., 2016). We use SAR DEMs from winter 2013/14 (Sommer et al., 2020) and 2018/19 to derive respective elevation mosaics. The 2018/19 TanDEM-X DEMs were created from ~160 Co-registered Single look Slant range Complex (CoSSC) acquisitions, using differential interferometry, and vertically and horizontally co-registered according to the workflow described by (Sommer et al., 2020). In this study, all elevation datasets are resampled to 30m grids.

## 2.4.2 Surface elevation changes

1969-2019 elevation changes of Austrian glaciers are inferred from the DHM69 and TanDEM-X. The TanDEM-X DEM and the DHM69 are vertically and horizontally co-registered, using non-glacierized and flat terrain (slope < 15°) outside glacierized




areas and water bodies (Braun et al., 2019). Eventually, the DEMs are differenced and elevation change rates ($\frac{\delta H}{\delta t}$) are calculated

from the TanDEM-X and DHM69 acquisition dates. Since the DHM69 was acquired between September and October 1969,

we use the average date (1. October 1969) as reference date. For the Swiss Alps, glacier elevation changes are obtained by

differencing the TanDEM-X 2018/19 DEM mosaic and the DHM25. As for the DHM69, the TanDEM-X DEM and the

DHM25 are vertically and horizontally co-registered to minimize elevation offsets. Thereafter, we use the individual glacier

reference years of the DHM25 (Fischer et al., 2015) (section 2.3.1) and the TanDEM-X acquisition dates to convert the height

difference into elevation change rates. The median observation period of all Swiss glaciers is 1975-2019 with glacier-specific

periods varying between 1961-2019 and 1991-2019. For the later reconstruction period (2000-2014), we use glacier elevation

change rates derived from differencing SRTM C-Band (February 2000) and TanDEM-X DEMs of winter 2013/14 (Sommer

et al., 2020). Data voids in the elevation change maps are in most cases caused by SAR layover and shadows. Therefore, we

apply a bilinear interpolation as recommended by a recent study (Seehaus et al., 2020). Finally, all elevation change fields are

consistently resampled to a spatial resolution of 30m.

The mean vertical precision of the glacier elevation change rate is derived as slope-dependant standard deviations on non-

glacierized areas. All elevation change values outside glacier areas are aggregated within 5° slope bins. Thereafter, standard

deviations of each slope bin are calculated and weighted by the respective total glacier area to derive the regionwide mean

uncertainty. Further details on the elevation change error calculation are described in (Sommer et al., 2020). For the DEM

differences of the DHM25, DHM69 and the 2019 TanDEM-X acquisitions, the mean regional elevation change uncertainty is

±0.26 ma[-1] and ±0.18 ma[-1], respectively. The slope-derived $\frac{\delta H}{\delta t}$ error of the period 2000-2014 is ±0.39 ma[-1] (Sommer et al.,

2020). Therefore, we use an average uncertainty of the elevation change fields of ±0.3 ma[-1]. It should be noted, that we did not

attempt a correction for height offsets between the optical/topographic and SAR DEMs due to signal penetration into the

glacier surface. Particularly, for the DEM-difference between the 2018/19 TanDEM-X DEMs of winter 2018/19 and DHM69

of autumn 1969, glacier surface elevations were acquired during different seasons and the presence of signal penetration is

likely for the TanDEM-X DEMs. However, we assume that the bias in elevation change due to radar penetration is small

because the snow cover of winter 2018/19 is probably partially invisible for the X-band SAR and the observation period is

very long (~40 years). In the case of the DHM25, no correction is applied because the exact mapping dates of the glacier areas

are unknown.

**2.4.3 Glacier outlines**

20[th] century outlines of Alpine glaciers are extracted from the 1969 Austrian (GI1) and 1973 Swiss glacier inventories

(SGI1973). The 1969 Austrian GI1 was originally compiled from the same aerial photographs as the DHM69 (Patzelt, 1980)

and later digitized (Lambrecht and Kuhn, 2007). Outlines of the SGI1973 are based on aerial photographs of September 1973

(Müller et al., 1976; Maisch et al., 2000) which were digitized and georeferenced by (Paul et al., 2004). Due to the varying

timestamps of the DHM25, many glacier outlines of the SGI1973 are older than the respective surface heights of the DHM25.



However, the glacier area change between 1973 and 1985 was very small (~ -1%) (Paul et al., 2004). Several glacier inventories covering the entire Alps have been created from satellite images and semiautomatic classification. The Randolph Glacier Inventory (RGI) (Pfeffer et al., 2014) of the European Alps was mostly created from band ratios of optical 2003 Landsat imagery with a resolution of 30 m (Paul et al., 2011). 2013-15 glacier extents were mapped from Landsat 8 images (Sommer et al., 2020). The most recent alpine-wide inventory is based on 10 m-resolution Sentinel S2 acquisitions (Paul et al., 2020)

from the years 2015-17. Additionally, recent regional inventories are available for the Austrian Alps 2015 (Buckel et al., 2018) and Swiss Alps 2013-18 (SGI2016) (Linsbauer et al., 2021), based on manual delineation of glacier outlines from high-resolution optical images and elevation models. For each glacier inventory, adjacent glacier boundaries are removed and the respective glaciers are merged into continuous geometries. Thereby, inconsistencies in ice thickness across divides and ridges are avoided (Fürst et al., 2017).

### 2.4.4 Ice thickness observations


Reference ice thickness observations of Alpine glaciers are available via the Glacier Thickness Database (GlaThiDa V.3) (GlaThiDa Consortium, 2020), which is a standardized collection of remote sensing & in-situ measurements, and a recent publication on helicopter-borne ground-penetrating radar (GPR) measurements of almost all larger Swiss glaciers (Grab et al., 2021). While the GlaThiDa database of Alpine glaciers includes a large number of observations from different time periods

and measurement techniques, the dataset by (Grab et al., 2021) provides mainly GPR tracks between 2016 and 2020 but also older, so far not publicly available, measurements. Most of the in-situ thickness observations are very densely spaced. Therefore, we removed observations which were less than 30 m apart, resulting in a total number of ~53,000 in-situ measurements. The mean measurement uncertainty of all in-situ observations is 8.2 m. For ~30% of the observations, the error is unknown because there is no information on the measurement uncertainty within the GlaThiDa database. For those

measurements, the uncertainty is approximated as 20% of the respective ice thickness value.

Thickness observations of the glacier margins are derived from glacier areas which became ice-free as delineated by the multi-temporal outline and elevation information. Therefore, absolute elevation change values ($\delta H$) are extracted at glacier retreat areas which were inferred from differencing respective glacier inventories. Additionally, an inner and outer buffer of 30 m is applied to the glacier retreat areas and a slope threshold of 25° to exclude values close to the glacier outlines or at steeper

slopes which tend to be less reliable. In summary, we derive approximately 140,000 thickness observations from glacier areas which became ice-free between 1969 (AT) and ~1970 (CH) and 2019. For the period 2000-2014, we obtain 70,000 observations. The uncertainty of the $\delta H$ measurements can be estimated from the errors of the $\frac{\delta H}{\delta t}$ fields (section 2.4.2.) and the respective observations periods of the DEM-differences. Hence, the mean vertical $\delta H$ uncertainties for the period ~1960-2019 are ±10.3 m and ±8.9 m for Swiss and Austrian glaciers and ±5.5 m for the period 2000-2014.





### 2.4.5 Surface mass balance

SMB estimates for all glaciers of the European Alps (referring to the RGI v6.0) are available from the Global Glacier Evolution Model (GloGEM, Huss and Hock, 2015; Zekollari et al., 2019). The model describes the main processes of mass balance – accumulation, melt and refreezing – and provides annual SMB for 10 m elevation bins between 1951 and 2019. The model was driven by the E-OBS dataset (Cornes et al., 2018) and has been calibrated to match glacier-specific mass changes 2000-2019 (Hugonnet et al., 2021). For the ice thickness reconstruction in this study, we averaged the annual SMB of each glacier over the periods 1969-2019 and 2000-2014, respectively. The elevation-binned SMB information is transferred to 30m grids using linear interpolation. To account for variations in glacier area and elevation over time, we use the SRTM DEM and RGI outlines for the period 2000-2014 and the Swiss and Austrian glacier inventories (GI1 & SGI1973) as well as surface elevations of the DHM69 and DHM25 for ~1970-2019. For the historic glacier inventories (GI1 & SGI1973), we spatially matched the glacier areas and the IDs of the RGI by comparing the respective outlines. In cases where the historic outline overlapped with more than one RGI ID, due to differences in the delineation of ice divides or the disintegration of a once continuous glacierized area into several smaller glaciers, we used the SMB values of the RGI geometry which had the largest spatial overlap. In addition, we had to apply hypsometric extrapolation in some cases to vertically extend the SMB information because the glacier outline of the historic inventories covered a larger elevation range than the respective RGI geometry.

### 2.5 Experimental setup

For the mass-conservation reconstruction, information on the glacier extent, surface topography and surface mass balance and elevation changes are required. Additional thickness observations are used to constrain the estimated ice thickness. Based on the above presented input datasets, we first calibrate the reconstruction method during the full observational period 1970-2019 in Switzerland and Austria. Then, the method is transferred to the entire European Alps focussing on the period 2000-2014. A detailed overview of the experimental setup and used input datasets is shown in Fig.1 and described in detail in the following sections.




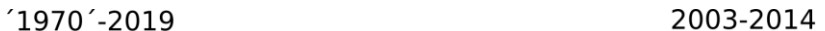

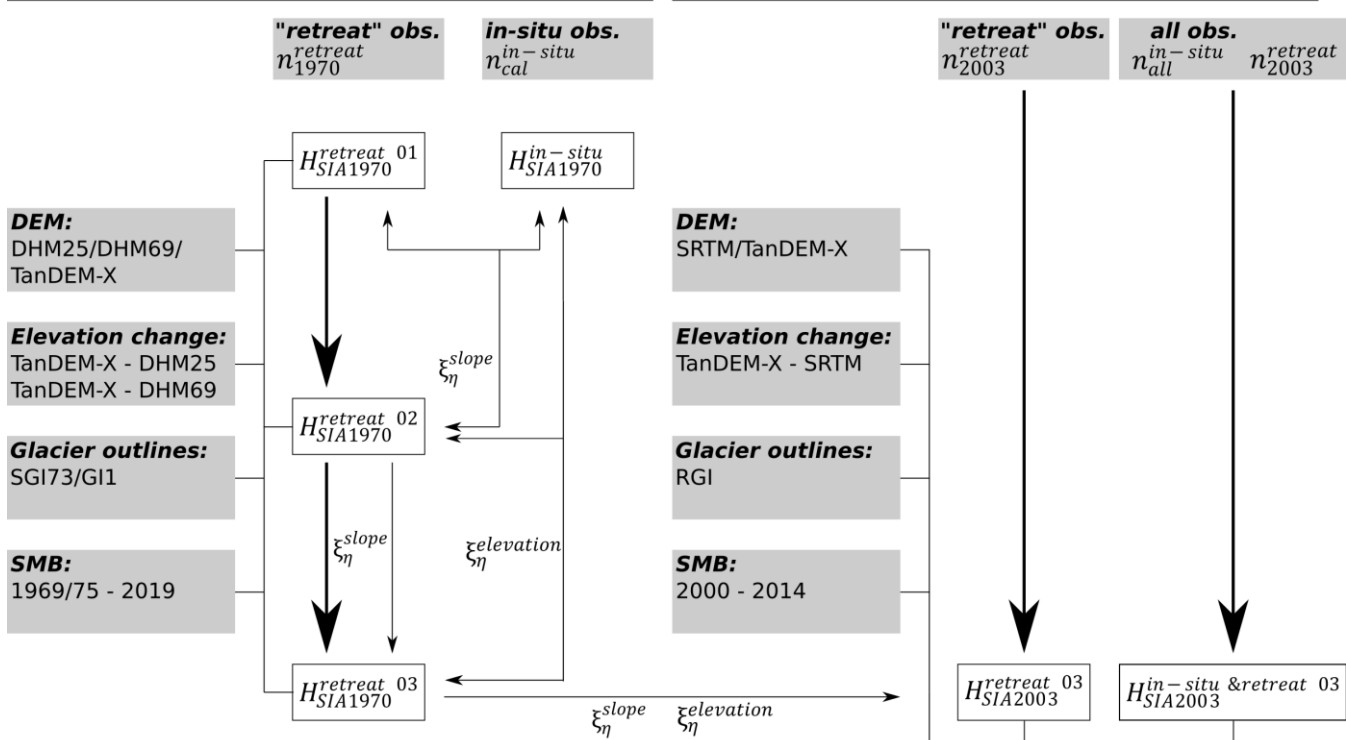

**Figure 1: Visualization of experimental setup and input datasets: Over the period ´1970´-2019, surface slope- and elevation-based scaling factors ($\xi_\eta^{slope}$ & $\xi_\eta^{elevation}$) are calibrated by comparing the ice thickness distribution ($H_{SIA1970}$) derived from in-situ thickness observations ($H_{SIA1970}^{in-situ}$) and retreat thickness observations at the glacier margins ($H_{SIA1970}^{retreat}$). By iterating $H_{SIA1970}^{retreat}$ ($H_{SIA1970}^{retreat\ 01}$ - $H_{SIA1970}^{retreat\ 03}$), $\xi_\eta^{slope}$ and $\xi_\eta^{elevation}$ are calibrated. Eventually, the calibrated scaling factors ($\xi_\eta^{slope}$ & $\xi_\eta^{elevation}$) are transferred to the period 2003-2014 to estimate the total alpine-wide glacier volume from different samples of ice thickness observations ($H_{SIA2003}^{in-situ\ \&\ retreat\ 03}$ & $H_{SIA2003}^{retreat\ 03}$).**

### 2.5.1 Reconstruction calibration 1970-2019

For the early period 1969-1973, glacier heights are extracted from the Swiss DHM25 and Austrian DHM69 while glacier areas are taken from the SGI1973 and GI1. The acquisition dates of the respective glacier outlines mostly refer to the years 1973 (Switzerland) and 1969 (Austria). Regarding the glacier surface topography, the Austrian DHM69 represents surface heights of the same year as the glacier inventory whereas the Swiss DHM25 elevations refer to regionally different acquisition years (see section 2.3.1). In the following sections, the reconstruction of the historic Swiss and Austrian ice thickness is therefore denoted as $H_{SIA1970}$ according to the approximate acquisition dates of the glacier outlines and DEMs ('1970'). As input fields for the $H_{SIA1970}$ reconstruction, elevation change rates are derived from the difference of the DHM25/DHM69 and TanDEM-



X mosaic of winter 2018/19. The $H_{SIA1970}$ is applied for two experimental setups, using different samples of thickness

observations and viscosity scaling factors:

For the initial reference run ($H_{SIA1970}^{in-situ}$), the available in-situ thickness observations (GlaThiDa Consortium, 2020; Grab et al.,

2021) of glaciers in the Swiss and Austrian Alps are divided into two equal sub-samples. All glaciers with in-situ measurements

(304 glaciers total) are grouped into four equal size classes by using the glacier-area quantiles. Thereafter, half of the glaciers

of each class are selected randomly to create a set of calibration and validation glaciers (152 glaciers each). Based on those

randomly selected glaciers, the in-situ thickness observations are divided into a set of calibration ($n_{cal}^{in-situ}$ = 24677

measurements) and validation ($n_{val}^{in-situ}$ = 25753 measurements) points. The thickness measurements of these observational

datasets are rather equally distributed across lower and upper sections of the glaciers and well represent both the thick and

central parts as well as the glacier margins. A detailed overview of the random selection of calibration and validation glaciers

is provided in Table S1 & S2 and Fig.S1 & S2. All in-situ thickness observations of the 152 calibration glaciers are used for

$H_{SIA1970}^{in-situ}$. No scaling of the ice viscosity is applied (Fig.1; Fig.2a).

The second calibration setup ($H_{SIA1970}^{retreat}$) is exclusively based on thickness observations from glacierized areas which became

ice-free between the 1970s and today ($n_{1970}^{retreat}$). Unlike the in-situ measurements, the majority of these thickness observations

are derived at lower elevations and close to the margin or terminus as those glacier parts typically show higher retreat rates

than the accumulation areas. For the $H_{SIA1970}^{retreat}$ setup we perform three iterations of the thickness reconstruction (Fig.1; Fig.2b-

d): (1) without ice viscosity scaling ($H_{SIA1970}^{retreat\ 01}$), (2) glacier surface slope-based viscosity scaling ($H_{SIA1970}^{retreat\ 02}$; see section 3.2)

and (3) glacier surface elevation and slope based viscosity scaling ($H_{SIA1970}^{retreat\ 03}$; section 3.3). For each iteration of $H_{SIA1970}^{retreat}$, the

reconstructed viscosity and thickness are compared to the respective $H_{SIA1970}^{in-situ}$ values. The initial iteration ($H_{SIA1970}^{retreat\ 01}$) includes

no additional scaling of the ice viscosity, similarly as $H_{SIA1970}^{in-situ}$. Based on the difference in viscosity between $H_{SIA1970}^{in-situ}$ and

$H_{SIA1970}^{retreat\ 01}$ (section 3.2), empirical slope-based scaling factors can be derived which are then applied to the viscosity estimation

during the second calibration iteration ($H_{SIA1970}^{retreat\ 02}$) according to Eq. (5):

(5)    $\xi_\eta^{slope} = y_\eta^{slope} \times \left( \alpha - \alpha_\eta^{tres} \right)$

Where $\xi_\eta^{slope}$ is the slope-based viscosity scaling factor and depends on the local surface slope ($\alpha$). Calibration parameters are

a slope gradient factor ($y_\eta^{slope}$; units per degree) and the respective slope threshold ($\alpha_\eta^{tres}$) beyond which the viscosity ratio

equals 1.

Additionally, the slope-based scaling of $H_{SIA1970}^{retreat\ 02}$ (Fig.2c) is extended with an elevation-based viscosity scaling ($H_{SIA1970}^{retreat\ 03}$)

to avoid unrealistically high thickness values in the upper glacier parts. As the vertical extents of alpine glaciers vary

significantly, the elevations of each continuous glacier area are normalized between the lowest ($h_{min}=1$) and highest glacier

elevation ($h_{max}=0$). An additional quantile filter is applied to the elevation range which defines the lowest and highest 2% of





elevation values to 1 and 0, respectively. By this means, we compensate for uncertainties in the glacier area delineation as the lowest and highest points of the glacier outline are often difficult to identify due to debris coverage or firn. The second scaling factor is applied based on the linear regression between the ice viscosity and the normalized glacier elevation range (Eq. (6)):

(6)    $\xi_\eta^{elevation} = 1.0 + y_\eta^{elevation} \times \left(\tilde{h} - \tilde{h}_\eta^{tres}\right)$

Where $\xi_\eta^{elevation}$ is the empirical elevation range-based correction, which is derived from the normalized local glacier elevation ($\tilde{h}$), the slope of the linear regression ($y_\eta^{elevation}$) and the vertical threshold ($\tilde{h}_\eta^{tres}$) where no correction is applied. By applying Eq. (6), $\eta$ is corrected and the final flux field and ice thickness (Fig.2d) is recalculated considering the slope and elevation 295    dependant viscosity ratios ($H_{SIA1970}^{retreat\ 03}$).

**2.5.2 Alpine-wide glacier volumes 2003**

For the early 21st century ('2003'), the glacier volume of all Alpine glaciers ($H_{SIA2003}^{retreat\ 03}$) is estimated based on RGI glacier areas, the SRTM DEM and surface elevation changes and mass balance data of the period 2000-2014. Retreat thickness observations are extracted from glacier areas which became ice-free since 2000 ($n_{2003}^{retreat}$). Additionally, $H_{SIA2003}^{in-situ\ \&\ retreat\ 03}$ 300    is derived from the same input data. However, all available in-situ thickness observations ($n_{all}^{in-situ}$) are integrated as well as observations from glacier areas which became ice-free since 2000 ($n_{2003}^{retreat}$). For both reconstructions, viscosity correction factors are transferred from the estimates of the 1970s Swiss and Austrian glacier volumes (Fig.1).











Figure 2: Reconstructed ice thickness of glaciers of the Bernese Alps (Jungfrau-Aletsch) for the 1970s: a) Glacier thickness estimated
from in-situ thickness measurements ($H^{in-situ}_{SIA1970}$). Locations of observations from field surveys are indicated as black triangles (Grab
et al., 2021). b) Glacier thickness based on thickness observations (black dots) from glacier retreat areas ($H^{retreat\ 01}_{SIA1970}$). c) Thickness
based on retreat observations and slope-dependant viscosity scaling ($H^{retreat\ 02}_{SIA1970}$). d) Thickness based on retreat observations and
slope- as well as elevation-dependant viscosity scaling ($H^{retreat\ 03}_{SIA1970}$).

## 3 Results

### 3.1 1970s ice thickness reconstruction and viscosity calibration

#### 3.1.1 In-situ thickness reconstruction

The '1970' reference ice thickness ($H_{SIA1970}^{in-situ}$) of Swiss and Austrian glaciers is estimated from the historic Swiss and Austrian
glacier inventories (SGI1973 & GI1), DEMs (DHM25 & DHM69) and respective surface elevation change and mass balance
data. In addition, all $n_{cal}^{in-situ}$ thickness observations are included to constrain the reconstructed ice thickness distribution. In
most cases the survey dates of the in-situ measurements differ from the acquisition dates of the DEMs. To derive the respective
ice thickness at the acquisition date of the DEM, the in-situ observations have to be temporally homogenized. Therefore, we
exclusively permit measurements that include both thickness and surface elevation and thus give information on the local basal
elevation beneath the glaciers. This is the case for all of the GPR thickness observations (Grab et al., 2021) and almost all of
the GlaThiDa entries. For the thickness homogenisation and the reconstructions, we assume negligible changes in the basal
elevation and subtract it from the reference DEM in ´1970´ and 2000. The estimated ice volume for the Swiss and Austrian
Alps ($V_{SIA1970}^{in-situ}$) is 125.4±24.7 km³, corresponding to a glacierized area of 1792.9 km², respectively. This is equivalent to 96%
of the total glacier area of the 1973 Swiss and 1969 Austrian inventory.

#### 3.1.2 Slope-based viscosity scaling

The initial retreat-areas reconstruction ($H_{SIA1970}^{retreat\ 01}$) is based on the same input data as the reference thickness ($H_{SIA1970}^{in-situ}$) but
instead of the in-situ observations, thickness values are extracted from deglacierized areas (section 2.4.4.). No temporal
homogenization of the observations has to be applied because the ice thickness is directly derived for the state of the reference
DEMs. Compared to the $H_{SIA1970}^{in-situ}$ reconstruction, the $H_{SIA1970}^{retreat\ 01}$ underestimates the total glacier volume ($V_{SIA1970}^{retreat\ 01}$) by
approximately 40% (Tab.1) due to a strong negative bias of the estimated ice thickness (Fig.3b). The largest differences are
found at the troughs of large valley glaciers where the observed ice thickness can be twice as high as the reconstructed thickness
(Fig.1). These observations are very similar to the "low elevation bias" configuration used in ITMIX2. The participating
models showed large deviations when the available thickness observations were limited to the low and thin glacier parts where
the ice flux is most likely underestimated due to substantial thinning rates and downwasting of the glacier termini (Farinotti et
al., 2021).

To estimate the bias in viscosity between $H_{SIA1970}^{retreat\ 01}$ and $H_{SIA1970}^{in-situ}$, viscosity values are extracted at locations with in-situ
observations ($n_{cal}^{in-situ}$) where the ice thickness and viscosity is known.  Viscosity values are then aggregated within 2° slope
bins to derive the ratio of $H_{SIA1970}^{retreat\ 01}$ and $H_{SIA1970}^{in-situ}$ viscosities (Fig.4). Average $H_{SIA1970}^{retreat\ 01}$ viscosities are in general lower than
$H_{SIA1970}^{in-situ}$ but the difference increases at slopes smaller than 25°. Using a linear regression, the slope-scaling parameters of Eq.
(5) are $y_{\eta}^{slope}$ = -0.08 and $\alpha_{\eta}^{tres}$ = 56.05°. The total ice volume ($V_{SIA1970}^{retreat\ 02}$) is 123.8±24.5 km³ (Tab.1).



**Figure 3: Differences between estimated ice thickness and observed ice thickness at locations of validation in-situ thickness measurements ($n_{val}^{in-situ}$ = 25753), root-mean-square-error (RMSE), standard deviation (Std.dev) and median difference (Median; all in metres) are stated in the lower right corner of each panel: a) reconstruction based on calibration ($n_{cal}^{in-situ}$) in-situ thickness measurements ($H_{SIA1970}^{in-situ}$) b) reconstruction based on all glacier thickness observations from deglacierized (retreat) areas ($H_{SIA1970}^{retreat\ 01}$), c) reconstruction based on all retreat observations and slope-based viscosity scaling ($H_{SIA1970}^{retreat\ 02}$) and d) reconstruction based on all retreat observations and slope- as well as elevation-based viscosity scaling ($H_{SIA1970}^{retreat\ 03}$).**

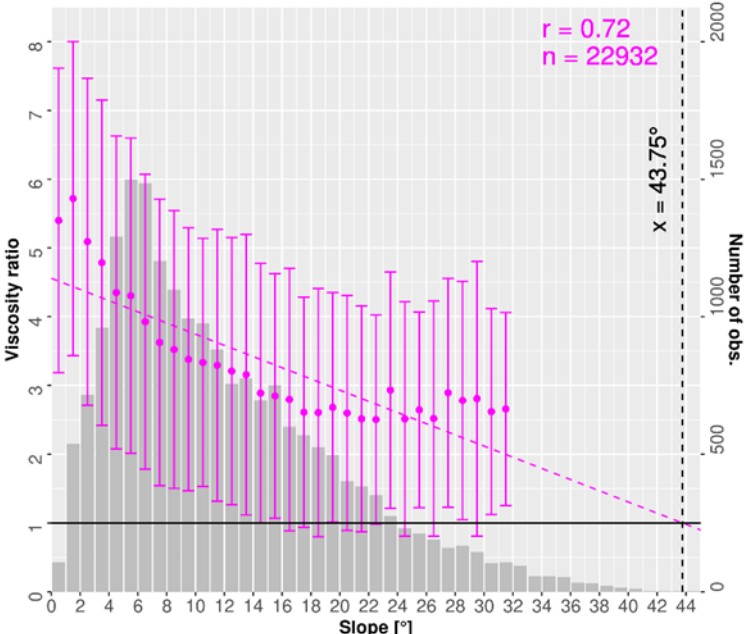

**Figure 4: Slope-dependant viscosity ratio of $H_{SIA1970}^{in-situ}$ and $H_{SIA1970}^{retreat\ 01}$ at locations of in-situ observations. Magenta points denote mean viscosity ratio values aggregated within 2° slope bins on glacierized areas. The number of observations of each slope bin is shown as gray bars. The respective linear regression of the slope-derived viscosity ratio is shown as dotted line. Ratios of elevation bins with less than 100 observations are excluded from the analysis. Vertical error bars indicate the respective mean ratio ± one standard deviation.**


### 3.1.3 Elevation-based viscosity scaling

The total ice volume $V_{SIA1970}^{retreat\ 02}$ is similar to $V_{SIA1970}^{in-situ}$. However, the modelled ice thickness tends to overestimate the observed glacier thickness at high altitudes while the lower glacier parts continue to remain too thin (Fig.2c). The overestimation at high altitudes is likely caused by large firn and ice areas with small surface slope where the corrected viscosity is overestimated.

Compared to $H_{SIA1970}^{retreat\ 01}$, the strong negative bias between estimated and observed ice thickness (Fig.3a) is reduced. Nevertheless, there is a remaining negative offset for glacier parts with an observed ice thickness of more than 300 m (Fig.3c). Fig.5 shows the mean viscosity ratio of $H_{SIA1970}^{retreat\ 02}$ and $H_{SIA1970}^{in-situ}$ versus the normalized glacier elevation range (section 2.5.1.). The ratio of $H_{SIA1970}^{retreat\ 02}$ and $H_{SIA1970}^{in-situ}$ is close to 1 at $0.5 - 0.6$ normalized elevation, which is approximately equal to the glacier median elevation. Yet a distinct offset is noticeable at lowest and highest elevations where the $H_{SIA1970}^{retreat\ 02}$ ice viscosity is

under- and overestimated, respectively. To compensate this viscosity biases, $y_\eta^{elevation}$ = -2.14 and $h_\eta^{tres}$ = 0.61 are applied based on the linear regression (Eq. (6)).

As shown in Fig.3d, the deviation between observed and estimated ice thickness of thick glacier parts (> 300m ice thickness) further decreases after applying Eq. (6). However, the median difference of $H_{SIA1970}^{retreat\ 03}$ between observed and estimated ice





thickness also increases by ~10 m compared to $H_{SIA1970}^{retreat\ 02}$, indicating a slight overestimation of the ice thickness by Eq. (6).

The reconstructed total glacier volumes of the different reconstruction steps are shown in Tab.1. While there is little difference in the modelled glacier volume of $H_{SIA1970}^{retreat\ 03}$ and $H_{SIA1970}^{in-situ}$ (~2%), the inclusion of the elevation-based scaling further improves the spatial thickness distribution (Fig.2d).

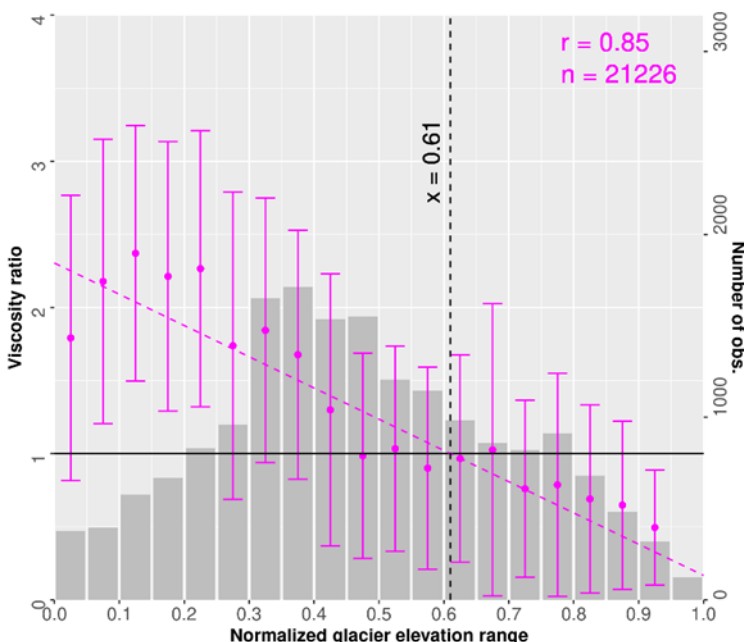

**Figure 5: Elevation-dependant viscosity ratio of $H_{SIA1970}^{in-situ}$ and $H_{SIA1970}^{retreat\ 02}$ reconstruction at locations of in-situ observations. Elevations have been normalized for each glacier with 0.0 being the minimum and 1.0 the maximum glacier height. Magenta points denote mean viscosity ratio values aggregated within 0.05 normalized elevation bins on glacierized areas. The number of observations of each bin is shown as gray bars. The respective linear regression of the elevation-derived viscosity ratio is shown as dotted line. Ratios of elevation bins with less than 100 observations are excluded from the analysis. Vertical error bars indicate the**
**respective mean ratio ± one standard deviation.**

### 3.2 2003 alpine-wide ice thickness reconstruction

$\xi_\eta^{slope}$ and $\xi_\eta^{elevation}$ are transferred to the early 21st century and the ice thickness of all Alpine glaciers is estimated based on the observation period 2000-2014. The estimated ice volume of $H_{SIA2003}^{in-situ\ \&\ retreat}$ is 124.8±23.5 km³. The volume of $H_{SIA2003}^{retreat}$

(134.2±40.3 km³) is relatively similar but ~7% higher, mainly due a likely overestimation of the glacier volume in the Austrian Alps (Tab.1).

In the following sections, these reconstructions are used to validate the estimated ice volumes against previous studies on Alpine glacier volumes (section 4.1) and compare the glacier-specific mean ice thickness of larger Alpine glaciers (section 4.2).

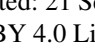




**Table 1: Overview of experimental setups for the reconstruction dates '1970' (Swiss & Austrian Alps) and '2003' (entire Alps) and estimated glacier volumes. $n_{obs}^{in-situ}$ and $n_{obs}^{retreat}$ indicate the number of thickness measurements from field surveys and DEM-differencing used in the respective experimental setup. Slope- and elevation-based viscosity scaling factors are given as $\xi_\eta^{slope}$ and $\xi_\eta^{elevation}$. Glacier areas refer to [1](Müller et al., 1976), [2](Patzelt, 1980) & [3](Paul et al., 2011)**

**\*Available in-situ thickness measurements in the Swiss & Austrian Alps were selected by a circular buffer with 30m distance and randomly grouped into two subsets.**

**\*\*The total volume of glaciers in CH/AT was reconstructed from all available thickness measurements while $y_\eta^{slope}$ and $y_\eta^{elevation}$ were derived from ~50% of all in-situ thickness measurements.**

| Experimental setup | Reconstruction date | Regions | Glacier area [km²] | $n_{obs}^{in-situ}$ | $n_{obs}^{retreat}$ | $\xi_B^{slope}$ | $\xi_B^{elevation}$ | $V_{SIA}$ [km³] |
|---|---|---|---|---|---|---|---|---|
| $H_{SIA1970}^{in-situ}$ | | | | 24677 / 25753 | - | - | - | 125.4±24.7 \*\* |
| $H_{SIA1970}^{retreat\,01}$ | '1970' | CH / AT | 1792.9[1,2] | - | | - | - | 72.2±19.1 |
| $H_{SIA1970}^{retreat\,02}$ | | | | - | 141482 | $y_\eta^{slope}$ | - | 122.5±24.5 |
| $H_{SIA1970}^{retreat\,03}$ | | | | - | | $y_\eta^{slope}$ | $y_\eta^{elevation}$ | 129.1±24.2 |
| $H_{SIA2003}^{retreat}$ | | Alps | 1997.6[3] | - | | | | 134.2±40.3 / 124.8±23.5 |
| $H_{SIA2003}^{in-situ\,\&\,retreat}$ | '2003' | FR | 195.5 | 53952 | 69022 | $y_\eta^{slope}$ | $y_\eta^{elevation}$ | 13.5±1.4 |
| | | CH | 1022.6 | | | | | 78.6±13.8 |
| | | AT | 355.9 | | | | | 13.3±3.8 |
| | | IT | 416.4 | | | | | 19.4±4.5 |





## 4 Discussion

### 4.1 Glacier volume comparison

The total '1970' ice volumes of the Swiss and Austrian Alps derived by this study are $H_{1970}^{in-situ} = 125.4\pm24.7$ km$^3$ or $H_{1970}^{retreat\ 03}$ = 129.1±24.2 km$^3$ (Tab.1). For the 1973 Swiss glacier inventory, the estimated volume is $V_{1970}^{in-situ} = 97.7\pm18.9$ km$^3$ or $V_{1970}^{retreat\ 03}$ = 99.5±16.9 km$^3$. Earlier estimates based on the same glacier area data, are available from a number of studies.

(Müller et al., 1976) and (Maisch et al., 2000) reported values of 67 km$^3$ and 74 km$^3$ using empirical relationships between glacier area and mean ice thickness. An ice volume of 75±22 km$^3$ was estimated by (Linsbauer et al., 2012) from a subset of glaciers with thickness observations and modeled ice thickness. Based on temporal extrapolation, an ice volume of 94.0±10.9 km$^3$ for the year 1973 was reported (Grab et al., 2021). While the ice volumes of the earlier studies (Müller et al., 1976; Maisch et al., 2000) are substantially lower than our estimate (~30-40%), good agreement is found with the estimate of the recent study

by (Grab et al., 2021) which supports their observation of a potential underestimation of 1970s glacier volumes by previous studies. For the estimate by (Linsbauer et al., 2012), their ice volume is also ~25% lower than this study yet the error bars overlap. For the Austrian Alps, our estimated ice volume is $V_{1970}^{in-situ} = 27.7\pm5.8$ km$^3$ or $V_{1970}^{retreat\ 03} = 28.7\pm7.3$ km$^3$ based on the 1969 glacier outlines. A ~20% lower ice volume (22.3 km$^3$) for the same year was calculated by (Helfricht et al., 2019) from a subset of thickness observations and a calibrated thickness model.

For the entire Alps, our estimated ice volume of the year '2003' is 124.8±23.5 km³ ($H_{SIA2003}^{in-situ\ \&\ retreat}$; $H_{SIA2003}^{retreat} = 134.2\pm40.3$ km³). Glacier volumes of the early 21$^{st}$ century were also reported for the entire Alps (Farinotti et al., 2019a; Millan et al., 2022), the Swiss Alps (Farinotti et al., 2009; Linsbauer et al., 2012; Grab et al., 2021) and Austrian Alps (Helfricht et al., 2019). An alpine-wide glacier volume of 130±30 km³ was calculated as consensus estimate from an ensemble of up to five models (Farinotti et al., 2019a) which is close to our estimate. A variant of the reconstruction approach (Fürst et al., 2017),

used here, also contributed to the consensus estimate. At the time, thickness observations were limited to GlaThiDa2.01 (Gärtner-Roer et al., 2014; Farinotti et al., 2019a) and thereby ignored the most recent and comprehensive measurements in Switzerland (Grab et al., 2021). An ice volume of 120±50.0 km³ (2017-2018) for the European Alps and Pyrenees was derived by a recent global study (Millan et al., 2022), based on flow velocity data, which is less than the consensus estimate results and the 2003 glacier volume of this study.

For Swiss glaciers, ice volumes of 74.9±9 km$^3$ (Farinotti et al., 2009), 65±20 km$^3$ (Linsbauer et al., 2012) and 77.2±4.6 km$^3$ (Grab et al., 2021) have been estimated for the beginning of the 21$^{st}$ century. The 2000 Swiss ice volume reconstructed from all available thickness observations ($H_{2003}^{in-situ\ \&\ retreat}$) of this study is 78.6±13.8 km³ which is ~20% higher than the estimate by (Linsbauer et al., 2012) but close to the glacier volume by (Farinotti et al., 2009) and (Grab et al., 2021). Nevertheless, the error bars of all estimates overlap. Regarding glaciers of the Austrian Alps, the estimated ice volume of 13.3±3.8 km$^3$ ('2003')

is lower than the value by (Helfricht et al., 2019) for the year 1998 (19.7 km$^3$). It is noteworthy that the glacierized areas of the estimates by (Farinotti et al., 2009; Linsbauer et al., 2012; Helfricht et al., 2019; Grab et al., 2021) and this study vary as





those studies used the respective regional glacier inventories of the Swiss and Austrian Alps. Differences in the delineation of glacier areas arise from the applied datasets and interpretation of glacier areas.

**4.2 Reconstructed ice thickness distribution**

To evaluate the ice thickness distribution of the $H_{SIA2003}^{retreat}$ reconstruction, the estimated ice thickness maps are directly compared to previous reconstructions of the 21$^{st}$ century glacier volume of the European Alps.

A comparison of previous ice thickness reconstructions based on different reconstruction approaches (Farinotti et al., 2019a; Helfricht et al., 2019; Grab et al., 2021; Millan et al., 2022) is shown in Fig.6 for Grosser Aletsch (Fig.6a-e) and Pasterze (Fig.6f-j). While the $H_{SIA2003}^{in-situ\,\&\,retreat}$ ice thickness maps are relatively similar to the other in-situ observations based

reconstructions, the estimated ice volume of the $H_{SIA2003}^{retreat}$ reconstruction is spread more evenly across the entire glacier domain, i.e. there is a tendency to overestimate or underestimate the thickness of thin and thick glacier parts, respectively. This is particularly noticeable in the upper areas of the Grosser Aletsch (Fig.6b). Occasionally, we find spuriously large values in certain confined areas, for instance in the upper part of Pasterze (Fig.6g). In contrast, glacier parts with very high ice thickness are often underestimated by the $H_{SIA2003}^{retreat}$ reconstruction such as Konkordiaplatz of the Grosser Aletsch (Fig.6b).

For a direct assessment of glacier-specific mean ice thickness, we refer to a comparison between published ice thickness maps of prominent Alpine glaciers (Farinotti et al., 2019a; Helfricht et al., 2019; Grab et al., 2021; Millan et al., 2022) and this study (Fig.7). For the comparison of mean glacier thicknesses, we use the respective glacier elevation change rates (section 2.4.2) to reduce temporal differences between the datasets. The reason is that previous glacier thickness maps refer to the years 2016-18 (Grab et al., 2021; Millan et al., 2022) and 2006 (Helfricht et al., 2019). Nevertheless, both values, the originally published

and temporally extrapolated mean ice thickness of each study, are shown in Fig.7. In general, the mean $H_{SIA2003}^{in-situ\,\&\,retreat}$ and $H_{SIA2003}^{retreat}$ ice thickness of most glaciers is similar as previously reported values. However, in some cases, the $H_{SIA2003}^{retreat}$ reconstruction deviates more from the mean glacier thickness found by other studies. Particularly for Adamello and Trift glacier, the mean ice thickness of the $H_{SIA2003}^{retreat}$ reconstruction is substantially lower or higher.

Point-specific offsets between in-situ measurements of ice thickness and reconstructed glacier thickness are shown in Fig.8.

Fig.8a,b refer to previously published alpine-wide ice thickness maps (Farinotti et al. (2019) and Millan et al. (2022). Differences between estimated and observed ice thickness in Fig.8a,b are derived from all in-situ observations (GlaThiDa; Grab et al 2021). Note that for $H_{SIA2003}^{retreat}$ differences are only computed on glaciers for which thickness measurements were ignored in the calibration (Fig.8c). These first three approaches have all been regionally calibrated in the Alps and do not necessarily reproduce local thickness observations. In this sense, a direct comparison is reasonable. The root-mean-square-

errors (RMSE) of local thickness are 55 m and 74 m for the datasets by (Farinotti et al., 2019a) and (Millan et al., 2022), while both studies indicate a general underestimation (26 m and 33 m median deviation, respectively). Thickness differences of the $H_{SIA2003}^{retreat}$ reconstruction indicate no obvious trend of an over- or underestimation of ice thickness for most glacier parts and the





median difference is significantly reduced with respect to the previous reconstructions. Concerning the standard deviation, the $H_{SIA2003}^{retreat}$ results are similar to both Farinotti et al. (2019a) and Millan et al. (2022).

Additionally, the ability of the applied reconstruction to reproduce available thickness observations is demonstrated in Fig. 8d. The underlying model by Fürst et al. (2017) is specifically constructed with a focus on the integration and reproduction of observed ice thicknesses. In this context, there are no indications of a systematic bias in ice thickness, introduced by the viscosity scaling approach and the retreat observations. Remaining deviations of about 10m are likely associated to the posteriori interpolation of the thickness map to the measurement locations, which is required for this comparison.

In summary, the $H_{SIA2003}^{retreat}$ ice thickness distribution appears to be smoother than reconstructions including in-situ measurements ($H_{SIA2003}^{in-situ \& retreat}$). This implies larger local uncertainties. To a certain degree, this has to be expected as the ice thickness of the inner glacier parts is naturally better constrained if respective field surveys are available. For the viscosity-scaling based reconstruction in contrast, the ice thickness of the inner glacier parts is widely unknown and has to be estimated exclusively from the viscosity correction parameters. Depending on the specific glacier morphology, this can result in large

local uncertainties (Fig. 6,7). Particularly, topographic characteristics of the glacier bedrock, such as overdeepenings or cirque thresholds, are challenging to reproduce when there are no direct observations of the thick glacier parts.





**Figure 6: Reconstructed ice thickness (H$_{SIA2003}^{in-situ\,\&\,retreat}$) of glaciers in the European Alps ('2003'): I) Mont Blanc Group, Pennine & Bernese Alps, II) Ötztal & Stubai Alps, III) Zillertal Alps, Venediger & Glockner Group, IV) Silvretta Alps. Comparison of estimated ice thickness distribution of Aletsch (CH) (a-e) and Pasterze Glacier (AT) (f-j) by this study H$_{SIA2003}^{in-situ\,\&\,retreat}$ (a,f) and H$_{SIA2003}^{retreat}$ (b,g) , (Farinotti et al., 2019a) (c,h), (Grab et al., 2021) (d), Helfricht et al 2019 (i) and (Millan et al., 2022) (e,j). The ice thickness maps by Farinotti et al. (2019a) refer to the results of the multi-model ensemble of ITMIX. The modeled ice thickness of Grosser Aletsch (Grab et al. 2021) and Pasterze (Helfricht el al. 2019) are constrained by in-situ measurements. Millan et al. (2022) uses glacier flow velocities from remote sensing data to derive the ice thickness distribution. (Background: SRTM hillshade).**

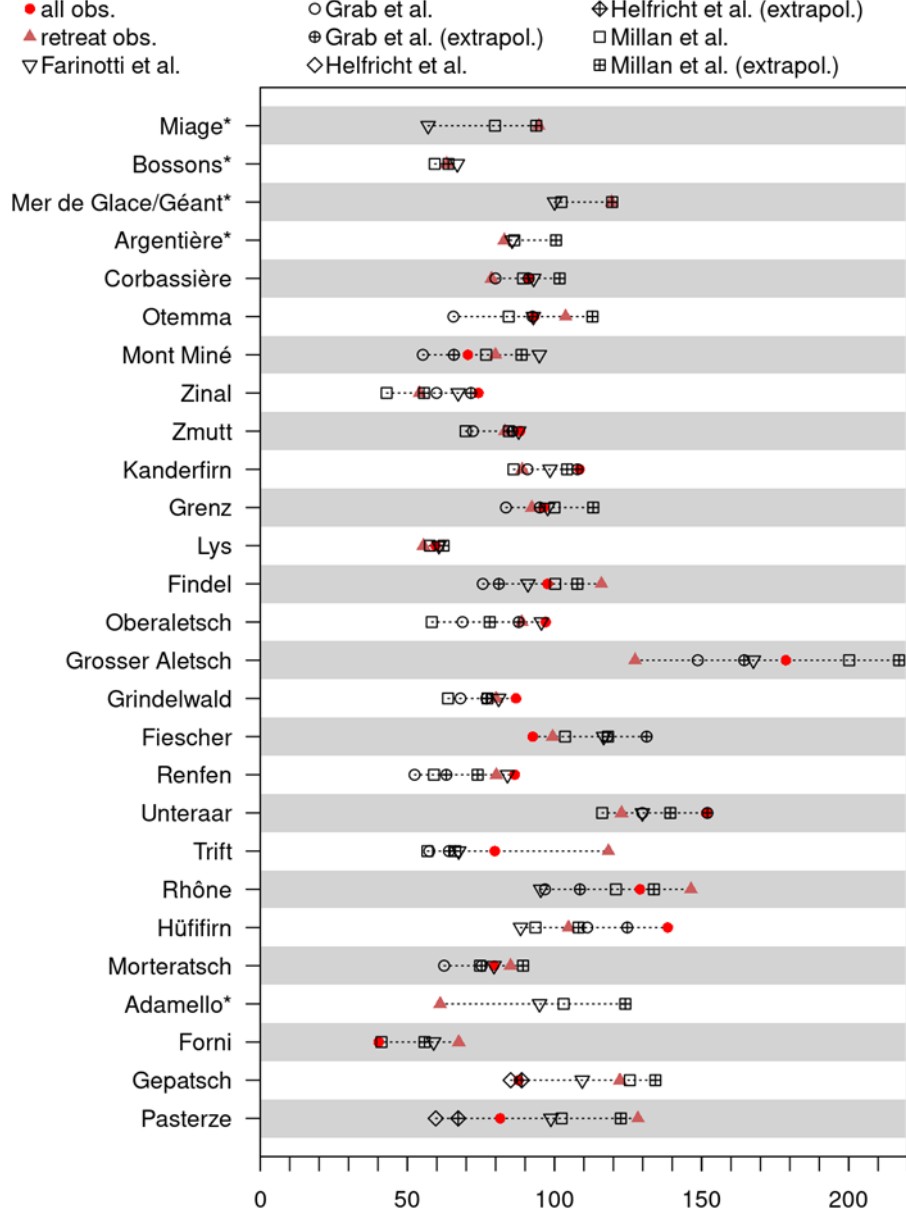

**Figure 7: Mean ice thickness of large Alpine glaciers (> 10 km²) estimated by this study and previous reconstructions (Farinotti et al., 2019a; Helfricht et al., 2019; Grab et al., 2021; Millan et al., 2022). Red dots and brown triangles indicate the mean $H^{\text{in−situ \& retreat}}_{\text{SIA2003}}$ and $H^{\text{retreat}}_{\text{SIA2003}}$ glacier thickness of this study. The reconstructions by (Helfricht et al., 2019), (Grab et al., 2021) and (Millan et al., 2022) refer to the years 2006, 2016 and 2017-18, respectively. Therefore, hollow symbols represent the original mean value derived from the published ice thickness maps while filled symbols show the mean glacier thickness temporally extrapolated to the year 2000. *For glaciers without in-situ observations, only the $H^{\text{retreat}}_{\text{SIA2003}}$ reconstruction is shown.**




**Figure 8: Difference between in-situ observed and estimated (2000) local ice thickness, root-mean-square-error (RMSE), standard deviation (Std.dev) and median difference (Median; all in metres) are stated in the lower right corner of each panel: Panel a) & b) refer to published ice thickness maps by Farinotti et al. (2019a) and Millan et al. (2022). Differences between estimated and observed ice thickness are derived from all available in-situ measurements (GlaThiDa; Grab et al. 2021). Deviations in ice thickness of the $H_{SIA2003}^{retreat}$ reconstruction of this study are shown in c). Note that in-situ observations from glaciers used to derive the viscosity scaling factors (section 2.5) are excluded from the comparison. Panel d) indicates the ice thickness distribution based on all available in-situ and retreat observations as well as the viscosity scaling factors ($H_{SIA2003}^{in-situ\ \&\ retreat}$). Note the logarithmic scaling of the thickness observations distribution. Darkblue areas indicate hexbins with more than 200 thickness measurement locations.**

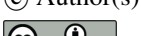


### 4.3 Uncertainty of viscosity scaling and retreat thickness

Similar as the findings of ITMIX2, a substantial underestimation of the glacier volume was found when relying solely on thickness observations from the lower and thin glacier parts. The presented viscosity scaling approach reproduces the regionwide glacier volume and thickness distribution well yet there are larger uncertainties in the glacier-specific ice thickness distribution.

Assuming the SIA, the ice thickness is derived from the glacier-wide flux field (Eq. (3)) while the ice viscosity is initially 510 unknown. The viscosity is estimated at locations with thickness observations and subsequently interpolated across the entire glacier domain. For the $H_{SIA1970}^{retreat}$ reconstructions, the interpolated viscosity field is in general lower than the $H_{SIA1970}^{in-situ}$ viscosity field as the used observations exclusively represent the relatively thin glacier margins with low viscosity values. Conversely, the $H_{SIA1970}^{in-situ}$ viscosity distribution is derived from thickness observations of the inner glacier parts with higher viscosity values, resulting in an overall higher ice thickness. To account for this generic limitation of the glacier margin-based viscosity 515 interpolation, the $H_{SIA1970}^{retreat}$ ice viscosities have to be corrected before the final ice flux is calculated.

The initial slope-dependant viscosity correction ($\xi_\eta^{slope}$) shows a tendency of overestimating the ice thickness at high altitudes. This pattern can be directly observed in the ice thickness maps (Fig.5). This is likely caused by the relatively large and flat accumulation areas of some Alpine glaciers where a high correction factor is applied to the ice flux. To compensate for this overestimation of ice thickness, the additional elevation-dependant scaling ($\xi_\eta^{elevation}$) reduces the correction factor in the 520 upper glacier parts and vice versa. However, the $\xi_\eta^{elevation}$ correction, which is based on the normalized glacier elevation range, can be biased when applied to large consecutive glacier areas. For instance, in the case of the Jungfrau-Aletsch Glacier area (Fig.1) the ice flux correction of the smaller glaciers is biased by the asymmetric distribution of the vertical extents of the individual glaciers. While the overall elevation range of this consecutive glacier area is determined by the terminus and high firn areas of the Grosser Aletsch, the adjacent glaciers (e.g. Oberaletschgletscher) have a much smaller vertical extent. This 525 can result in rather high or low $\xi_\eta^{elevation}$ correction factors depending on the vertical extent of the glacier in relation to the entire glacierized area. Another source of uncertainty can be an indeterminate separation between glacier ice and perennial snow as frequently found at high altitudes. In those cases, high thickness values are modeled on adjacent snow areas with small slopes and the normalized elevation range, which is used for the second correction step, can be shifted upwards resulting in rather thick accumulation areas.

In addition, it should be noted that the derived retreat thickness observations can be somewhat biased by terrain elevation changes in the glacier foreland, such as erosion and sedimentation, after the deglaciation. To reduce potential biases due to "unstable" height change measurements on glacier retreat areas, we exclude observations which are close (< 30 m) to either the past or present glacier outline and apply a slope threshold (25°). However, based on the available DEM-differences, it is not possible to differentiate between height changes prior to or after the deglaciation of the glacier foreland. Therefore, we 535 cannot completely avoid uncertainties of the extracted retreat thickness due to geomorphological processes.

All in all, the glacier-specific accuracy of the correction parameters and retreat thickness information can be somewhat influenced by the quality of the glacier inventory or the geometries of nearby glaciers. The approach is most favourable in cases where no or only a small sample of direct thickness observations is available. With the increasing number of satellite remote sensing data, the accuracy of the estimated ice thickness distribution can be improved by new high-resolution glacier
inventories or elevation change measurements. Eventually, the presented approach could be most beneficial in regions with large glacierized areas and sparse thickness observations where the glacier volume has to be interfered mostly from remote sensing information.

**5 Conclusions**

We present a topography-based scaling approach to estimate regionwide glacier volumes from retreat thickness observations
derived from remote sensing acquisitions. The method is based on an empirical relationship between in-situ observed and modelled ice thickness distributions of Alpine glaciers. Firstly, a slope-depending correction is applied to compensate for a general bias in the estimated ice volume due to the spatial distribution of retreat thickness observations. Secondly, an elevation-based correction is required to constrain the ice thickness distribution over the vertical glacier extent. It is shown that the applied viscosity corrections are able to provide a robust estimation of regionwide glacier volumes. Moreover, the empirical
scaling relations improve the distribution of the ice-thickness over the drainage basin. Compared to previous reconstructions and in-situ measurements of ice thickness, the median deviation between observed and modelled glacier thickness is significantly reduced (-6.3 m) whereas the root-mean-square error (60.8 m) and standard deviation (39.5 m) are similar. Yet, we still notice a tendency for thickness underestimation along the lower trunks and an overestimation at high altitudes where the topography is gently-sloping.
We provide additional ice thickness maps for '1970' (Swiss & Austrian Alps) and '2003' (Alps) which are derived from all available in-situ thickness measurements and retreat thickness values from DEM-differencing. The hereby reconstructed ice volume of the 21$^{st}$ century is 126.1±23.5 km³ ('2003') for the entire Alps. For glaciers of the Swiss and Austrian Alps, the ice volume was 125.4±24.7 km³ in '1970' and 93.9±17.6 km³ in '2003'. This implies a glacier volume loss of 0.8 %a$^{-1}$ (1.1 km³a$^{-1}$) over the period '1970'-'2003' in the Swiss and Austrian Alps.
The reconstruction approach shown in this study has the potential to constrain estimates of the ice thickness distribution in regions without direct observations of glacier thickness. Nevertheless, there is still room for improvements regarding the spatial distribution of estimated ice thicknesses. Particularly, the second elevation-based viscosity correction does not completely compensate for hypsometric biases in local ice thickness distribution in the case of certain glacier geometries. Furthermore, while the extraction of retreat ice thickness information from deglacierized areas is relatively straightforward in most mountain
regions, the transferability of the viscosity scaling factors derived from Alpine glaciers to other unsurveyed mountain regions needs to be assessed. Therefore, future work will have to address the applicability of the presented approach in regions with different glacier geometries and climatic settings, such as the South American Andes or High Mountain Asia.





**Acknowledgements**

This research was financially supported within the DFG Priority Program "Regional Sea Level Rise and Society" by the grant no. BR2105/14-2. J. Fürst has received funding from the European Union's Horizon 2020 research and innovation programme via the European Research Council (ERC) as a Starting Grant (StG) under grant agreement No 948290. We would like to thank M. Stocker-Waldhuber and M. Kuhn who provided access to the DHM69 dataset. The original version of the DHM25lvl1 was made available by swisstopo. TanDEM-X data were kindly provided free of charge by the German Aerospace Center (DLR)

under AO mabra_XTI_GLAC0264. Further, we would like to thank the GlaThiDa consortium (GlaThiDa Consortium, 2020) and M. Grab and co-authors (SwissGlacierThickness-R2020 https://doi.org/10.3929/ethz-b-000434697; Grab et al., (2021)) for making the unique database of in-situ observations of glaciers in the European Alps publicly available. The authors gratefully acknowledge the scientific support and HPC resources provided by the Erlangen National High Performance Computing Center (NHR@FAU) of the Friedrich-Alexander-Universität Erlangen-Nürnberg (FAU). NHR funding is provided

by federal and Bavarian state authorities. NHR@FAU hardware is partially funded by the German Research Foundation (DFG) – 440719683.

**Disclaimer**

The presented content only reflects the authors' views and the European Research Council Executive Agency is not responsible
for any use that may be made of the information it contains.

**Author contributions**

CS derived the glacier volume scaling factors, prepared all input datasets and wrote the manuscript. The applied ice thickness model was developed by JF. MH provided the surface mass balance data and aided with the experimental setup and
interpretation of results. JF and MHB initiated and led the study. All authors revised the paper.





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
