# Peer review of "Constraining regional glacier reconstructions using past ice thickness of deglaciating areas – a case study in the European Alps"

_The Cryosphere, 2022_

## Referee Comment (RC2)

**Review of Sommer et al. (2022): Constraining regional glacier reconstructions using past ice thickness of deglaciating areas – a case study in the European Alps**

**Summary**

This paper presents a new approach for reconstructing the thickness of glaciers using an additional constraint of DEM-differencing of areas that have become deglaciated within the period of observational record, building on previous work by the authors. They apply this method to the European Alps, reconstructing ice volumes in 1970 and 2003, and show that, deglaciation preferentially occurring in thinly glaciated areas, this induces a global bias in their results similar to that found in the ITMIX2 experiments when using thickness observations from thinner parts of glaciers. They consequently derive two different empirical correction factors for the modelled ice viscosity, one based on distance from the glacier margin and one on elevation, and show that including these substantially improves their reconstruction and leads to better matches between modelled and observed thicknesses than in previous studies, though still with possible large local mismatches owing to the regional-scale calibration of the correction factors. Overall, they find glacier volumes in-line with previous recent studies of Alpine glacier volume, but with the major advantage that their approach could be easily extended to areas without direct thickness observations.

All in all, I think this is a good paper with much to recommend it. Most of my concerns are of a fairly minor nature, though with a single larger one to consider. I feel the paper makes a valuable addition to the literature on estimating glacier thickness and volumes by proposing an innovative method that could be of use globally.

**Samuel**

**Major points**

• Applicability to other regions: The authors touch on this briefly at the end of the conclusion, but perhaps soft-pedal this problem a little too much to be considered entirely honest about it. The paper already shows that the large-scale calibration of the empirical factors leads to substantial local variation within the Alps; I thus can't help feeling that it's exceptionally unlikely that the same correction factors would work in an extra-Alpine context. I think some additional consideration of the challenge likely to be posed by trying to apply the method elsewhere needs to be included – nothing much, just another couple of sentences in the conclusion – but the current formulation is unrealistically optimistic, I feel.

**Minor points**

- p.1, l.13: I might say 'due to the difficulty of undertaking field surveys' or 'challenging field conditions'. Strictly speaking 'challenging field surveys' doesn't really mean what you're trying to get across it implies the surveys were difficult but have been done, when what you're trying to say is that they're difficult and therefore haven't been done.
- p.1, l.25: I'm wondering whether the reconstructed volume in 1970 is really the right thing for the abstract when you also calculate a modern glacier volume for the Alps. My feeling is that people will be more interested in the modern value and how it stacks up to other recent reconstructions of Alpine glacier volume, or the rate of change between the two periods you've reconstructed, than the volume 50 years ago so I'd suggest re-writing the abstract along those lines, certainly if you've only got space for one highlight result.
- p.2, 1.35: You need to spell out what GLOF stands for before using the abbreviation.
- p.2, l.50: 'Contrastingly'
- p.4, Eq. 2: I can't see η used anywhere in the equation, though the text (lines 101-103) implies it should be? As a result, I'm unclear exactly how your viscosity scaling is actually being applied to the flux field to modify the inferred ice thicknesses.
- p.6, l.165: 'slope-dependent'
- p.9, Fig. 1: I'm not entirely sure this figure helps explain things all that well. All the bidirectional arrows make it very challenging to work out where to start and re-reading the caption several times hasn't helped me make a lot more sense of it. I am prepared to accept that I'm not very good at

understanding diagrams, but if you can come up with a more intuitive schematic, that might not hurt. It made sense after I read Section 2.5.1, but not till then, so at the very least move it to after that section of text.

- p.11, l.295: 'elevation-dependent'
- p.12, Fig. 2: As a general point, 'dependant' is a word in English, but it's the noun form, so a dependant would be, say, your child. If you're aiming for the adjective, it's always 'dependent'. I'll stop pointing it out now, but go through and replace all instances of it (you almost certainly do not mean 'dependant' anywhere in the paper).
- p.15, Fig. 4: I'm not sure a linear regression is all that great a fit, based on the graph. It overestimates in the middle and underestimates at both extremes. I realise this is what the second correction factor is ultimately fixing, but is there any way you could test the impact of using a non-linear regression? Also, what is the dotted black line on the graph showing? I assume it's αtres, but then it's got a different value to that quoted in the text at l. 340. Please clarify what's going on here.
- p.16, Fig.5: Similarly to Fig. 4, you need to explain in the caption what the dotted black line represents. Here, the value matches up with the value quoted at l. 365 for htres, so I'm confident it's that, but it needs stating in the caption.
- p.18, Sect. 4.1: I'm wondering if you could be a little stronger here in your assertion that earlier studies might have underestimated Alpine glacier volumes. Given nearly all the recent studies are pointing in that direction, it seems to me that one would have to be extremely perverse to argue that the earlier studies weren't underestimates.
- p.25, l.541: 'inferred' not 'interfered'
- p.25, l.546: 'slope-dependent', not 'slope-depending'

---

## Author Comment (AC1)

**Author responses to review (02) of: *Constraining regional glacier reconstructions using past ice thickness of deglaciating areas – a case study in the European Alps**

*First of all, we would like to thank the reviewer for the valuable and constructive comments on our manuscript! The comments clearly improve the quality of this study and all comments are considered.*

*According to the points raised below, we extended the discussion of uncertainties regarding the transferability of the presented approach to other mountain regions (section 4.3). Furthermore, we made some changes to the presentation of results and visualization of the experimental workflow as suggested by the referee.*

*Our point-by-point responses are denoted below in bold. New and revised paragraphs which were included in the main manuscript are additionally indicated by the respective line numbers and bold italic text.*

**Referee #2**

Review of Sommer et al. (2022): Constraining regional glacier reconstructions using past ice thickness of deglaciating areas – a case study in the European Alps

**Summary**
This paper presents a new approach for reconstructing the thickness of glaciers using an additional constraint of DEM-differencing of areas that have become deglaciated within the period of observational record, building on previous work by the authors. They apply this method to the European Alps, reconstructing ice volumes in 1970 and 2003, and show that, deglaciation preferentially occurring in thinly glaciated areas, this induces a global bias in their results similar to that found in the ITMIX2 experiments when using thickness observations from thinner parts of glaciers. They consequently derive two different empirical correction factors for the modelled ice viscosity, one based on distance from the glacier margin and one on elevation and show that including these substantially improves their reconstruction and leads to better matches between modelled and observed thicknesses than in previous studies, though still with possible large local mismatches owing to the regional-scale calibration of the correction factors. Overall, they find glacier volumes in-line with previous recent studies of Alpine glacier volume, but with the major advantage that their approach could be easily extended to areas without direct thickness observations.

All in all, I think this is a good paper with much to recommend it. Most of my concerns are of a fairly minor nature, though with a single larger one to consider. I feel the paper makes a valuable addition to the literature on estimating glacier thickness and volumes by proposing an innovative method that could be of use globally.

Samuel

**Thank you very much for taking the time to review our manuscript. The comments significantly helped to improve the presentation & discussion of results in this study as well as the general legibility.**

**Major points**

**RC.02.01:** Applicability to other regions: The authors touch on this briefly at the end of the conclusion, but perhaps soft-pedal this problem a little too much to be considered entirely honest about it. The paper already shows that the large-scale calibration of the empirical factors leads to substantial local variation within the Alps; I thus can't help feeling that it's exceptionally unlikely that the same correction factors would work in an extra-Alpine context. I think some additional consideration of the challenge likely to be posed by trying to apply the method elsewhere needs to be included – nothing much, just another couple of sentences in the conclusion – but the current formulation is unrealistically optimistic, I feel.

> **Response: We agree with the reviewer that this is an important point. The unclear transferability of the approach to other regions is certainly one of the largest draw-backs regarding its broader application. Admittedly, the presented correction functions are somehow inherently linked to the geometries of the Alpine glaciers and respective distribution of small, medium and large glaciers. Unfortunately, we cannot entirely avoid this problem with the second experiment (~2000 ice volume reconstruction) because the overall geometries of the Alpine glaciers remain similar between the 1970s and 2000 (although glacier areas are retreating).**
>
> **Therefore, we extended the last part of the discussion section (4.3, l540) by:**
>
> *"Eventually, the presented approach could be most beneficial in regions with large glacierized areas and sparse thickness observations where the glacier volume has to be interfered mostly from remote sensing information. However, another potential source of uncertainty, regarding the transferability of the presented correction terms to glacierized areas outside the European Alps, results from the varying regional glacier morphologies in terms of size composition and elevation range. While the found empirical relations between ice viscosity and glacier surface topography have been applied to a different observation period and larger study region ($H_{SIA2003}^{retreat}$), we expect that the scaling functions are to some degree related to the geometries and size distribution of glaciers in the Swiss and Austrian Alps. In the European Alps, this uncertainty cannot be avoided because the overall distribution of a large number of small to medium-sized cirque glaciers with few large valley glaciers remains unchanged between $H_{SIA1970}^{retreat}$ and $H_{SIA2003}^{retreat}$, despite the substantial reduction in glacierized area since the 1970s. Further, the presented relations might be linked to the geographic environment of the European Alps as glacier changes are connected to the surrounding topography and climatic conditions (Abermann et al., 2011). To quantify these relations between the Alpine topography, glacier geometries and the derived scaling parameters and examine the transferability, it would be mandatory to extend the presented analysis to another glacierized region with different glacier morphology, such as marine- and lake-terminating glaciers, as well as different climatic settings, which is beyond the scope of this work."*

**Minor points**

**RC.02.02:** p.1, l.13: I might say 'due to the difficulty of undertaking field surveys' or 'challenging field conditions'. Strictly speaking 'challenging field surveys' doesn't really mean what you're trying to get across – it implies the surveys were difficult but have been done, when what you're trying to say is that they're difficult and therefore haven't been done.

> **Response: Agree. We changed this part accordingly.**

**RC.02.03:** p.1, l.25: I'm wondering whether the reconstructed volume in 1970 is really the right thing for the abstract when you also calculate a modern glacier volume for the Alps. My feeling is that people will be more interested in the modern value and how it stacks up to other recent reconstructions of Alpine glacier volume, or the rate of change between the two periods you've reconstructed, than the volume 50 years ago so I'd suggest re-writing the abstract along those lines, certainly if you've only got space for one highlight result.

> **Response: In the abstract, we did not include the volume of all Alpine glaciers for the year ~2000 because a number of previous studies have already presented volume reconstructions for the entire Alps. Therefore, our alpine-wide results should be rather seen as a proof of concept, by reproducing similar ice thickness as the reference studies (discussion section), but not as an entirely new result.**
> **As the abstract is already slightly too long, we removed the last sentence (and the 1970s volume) entirely from the abstract (l.25-26). Thereby, the abstract emphasizes the main outcome of this study, i.e. the approach of using remote sensing data and not the calculation of new glacier volume results for the European Alps.**

**RC.02.04:** p.2, l.35: You need to spell out what GLOF stands for before using the abbreviation.

> **Response: Yes, we inserted "glacial lake outburst floods" at this point**

**RC.02.05:** p.2, l.50: 'Contrastingly'

> **Response: We replaced "contrasting" by "contrastingly" in l.50.**

**RC.02.06:** p.4, Eq. 2: I can't see η used anywhere in the equation, though the text (lines 101-103) implies it should be? As a result, I'm unclear exactly how your viscosity scaling is actually being applied to the flux field to modify the inferred ice thicknesses.

> **Response: There is an error in the equation. "B" in Eq.2 should be "η" for ice viscosity. The reason is that we used "B" for viscosity in the first manuscript version but replaced it later by "η" because "B" is often used for mass balance and might be therefore confusing for some readers. In any case, we replaced "B" with "η" in Eq.2.**

**RC.02.07:** p.6, l.165: 'slope-dependent'

> **Response: Yes, we changed this here and in the rest of the paper**

**RC.02.08:** p.9, Fig. 1: I'm not entirely sure this figure helps explain things all that well. All the bidirectional arrows make it very challenging to work out where to start and re-reading the caption several times hasn't helped me make a lot more sense of it. I am prepared to accept that I'm not very good at understanding diagrams, but if you can come up with a more intuitive schematic, that might not hurt.

It made sense after I read Section 2.5.1, but not till then, so at the very least move it to after that section of text.

> **Response: We included this workflow figure in the paper in the attempt of presenting the individual stages (and structure) of the reconstruction visually but we agree that there is still a lot of room for improvements. We moved the figure at the end of the methods section (after section 2.5.2, p.11) and improved the general layout by using different color schemes (for different types of input data) and stroke widths. Additionally, the bidirectional arrows were replaced by one-directional arrows.**

**RC.02.09:** p.11, l.295: 'elevation-dependent'

> **Response: See comment above & below, we changed this throughout the entire manuscript.**

**RC.02.10:** p.12, Fig. 2: As a general point, 'dependant' is a word in English, but it's the noun form, so a dependant would be, say, your child. If you're aiming for the adjective, it's always 'dependent'. I'll stop pointing it out now, but go through and replace all instances of it (you almost certainly do not mean 'dependant' anywhere in the paper).

> **Response: Yes, we were obviously aiming for "dependent" and will correct this during the revision.**

**RC.02.11:** p.15, Fig. 4: I'm not sure a linear regression is all that great a fit, based on the graph. It overestimates in the middle and underestimates at both extremes. I realise this is what the second correction factor is ultimately fixing, but is there any way you could test the impact of using a non-linear regression? Also, what is the dotted black line on the graph showing? I assume it's αtres, but then it's got a different value to that quoted in the text at l. 340. Please clarify what's going on here.

> **Response: In Figure 4, we deliberately applied a rather simple linear regression because the analyzed ratio of surface slope and viscosity is not a physical-based but empirical relation found for flux estimation from the different subsets of thickness observations and glacier morphologies. Therefore, we did not attempt to apply a more complex non-linear scaling of the ice viscosity because the glacier-specific correction terms are likely more difficult to be controlled. Within the Alps, glacier geometries strongly vary between large valley glaciers and relatively small cirque glaciers, i.e. the extent or hypsometric location of flat or steep glacier parts varies significantly between individual glaciers. As described in the discussion, there is (for example) already a noticeable overestimation of very flat glacier parts. With a non-linear function this bias would be likely to even increase because the scaling ratio for very flat glacier areas would be even higher.**
>
> **→ Regarding the black dotted line shown in the graph at 43.75°, this is in fact the slope threshold of the linear regression ($\alpha_\eta^{thres}$). Unfortunately, there was an error in the text at L.340 / P.13. $\alpha_\eta^{thres}$ should be 43.75°, as indicated in figure 4, and not 56.05°. The latter number is the location where the slope viscosity ratio is equal 0. However, we use the surface slope (43.75°) where the ratio becomes 1 as threshold and do not apply the correction function to steeper slopes (where almost no glacier areas are located). For the revised manuscript, we replaced $\alpha_\eta^{thres}$ in L.340 with the correct value and extended the caption of Fig.4 by adding: *"The slope threshold ($\alpha_\eta^{thres}$) is indicated as vertical black dotted line at 43.75°."***

**RC.02.12:** p.16, Fig.5: Similarly to Fig. 4, you need to explain in the caption what the dotted black line represents. Here, the value matches up with the value quoted at l. 365 for htres, so I'm confident it's that, but it needs stating in the caption.

**Response: See comment above, the dotted black line represents the elevation threshold ($h_\eta^{tres}$). We will include this as description in the caption and replace the "x = 0.61" by "$h_\eta^{tres}$ = 0.61" in the figure.**

**RC.02.13:** p.18, Sect. 4.1: I'm wondering if you could be a little stronger here in your assertion that earlier studies might have underestimated Alpine glacier volumes. Given nearly all the recent studies are pointing in that direction, it seems to me that one would have to be extremely perverse to argue that the earlier studies weren't underestimates.

**Response: We agree with the reviewer that an underestimation of glacier volume by the early studies (Müller et al., 1976; Maisch et al., 2000) is likely. We rephrased this part during revision.**

**RC.02.14:** p.25, l.541: 'inferred' not 'interfered'

**Response: "Interfered" was replaced by "inferred".**

**RC.02.15:** p.25, l.546: 'slope-dependent', not 'slope-depending'

**Response: We inserted "slope-dependent".**

**References**

Abermann, J., Kuhn, M., and Fischer, A.: Climatic controls of glacier distribution and glacier changes in Austria, Ann. Glaciol., 52, 83–90, https://doi.org/10.3189/172756411799096222, 2011.

Maisch, M., Wipf, A., Denneler, B., Battaglia, J., and Benz, C.: Die Gletscher der Schweizer Alpen: Gletscherhochstand 1850, Aktuelle Vergletscherung, Gletscherschwund-Szenarien. (Schlussbericht NFP 31), 2nd ed., vdf Hochschulverlag an der ETH, Zürich, 373 pp., 2000.

Müller, F., Calflisch, T., and Müller, G.: Firn und Eis der Schweizer Alpen (Gletscherinventar), Geographisches Institut, ETH, Zürich, 1976.

---

## Author Comment (AC2)

**Author responses to review (01) of: *Constraining regional glacier reconstructions using past ice thickness of deglaciating areas – a case study in the European Alps**

*First of all, we would like to thank the reviewer for the valuable and constructive comments on our manuscript! The comments clearly improve the quality of this study and all comments are considered.*

*According to the points raised below, we addressed the differences in glacier volume between this study and previous calculations by estimating the geodetic volume change of Austrian Alps directly from the input DEMs and investigating technical differences in the delineation of glaciers from different inventories. We also revised the calculation of the regional glacier volume of the Austrian Alps for the year 1969. Furthermore, we extended the discussion of local biases in reconstructed ice thickness due to the regionally calibrated scaling functions and general uncertainties regarding the transferability of the approach to other mountain regions (section 4.3). Finally, we partially changed the illustration of thickness differences between this study and previous studies in Fig. 6, as suggested by the referee.*

*Our point-by-point responses are denoted below in bold. New and revised paragraphs which were included in the main manuscript are additionally indicated by the respective line numbers and bold italic text.*

**Referee #1**

In their manuscript „ Constraining regional glacier reconstructions using past ice thickness of deglaciating areas – a case study in the European Alps" C. Sommer et al. present a method to assimilate spatially distributed data of observed ice thickness loss at glacier tongues into regional calibration of ice thickness models.

Ice thickness observations in deglaciating areas are not necessarily representative for the state of the entire glacier. In particular at times of negative glacier disequilibrium, a direct assimilation for model calibration would lead to underestimation of the glacier volumes. Thus, they developed an empirical relationship between the ice viscosity at locations with in-situ observations and observations from DEM-differencing at the glacier margins to overcome this bias.

They authors combine ice thickness data sets, remote sensing products and modelling for glaciers cross the European Alps. With respect to data availability, the calibrated model is able to reproduce regional glacier volumes, but larger uncertainties remain at a local scale.

Nevertheless, the presented approach might be advantageous for improved estimation of glacier volumes if applied to regions where no direct measurements of glacier ice thickness exist.

However, the period of interest is in the manuscript comprises both states of glacier disequilibrium, with increasing glacier volumes in the 1970s and 80s, followed by years of strong negative glacier mass balances. This might be a challenging process to be considered in the discussion of the results.

In general, the manuscript is well written. The extensive introduction is followed by a well-structured data and methods section. The results are discussed in the light of existing literature and conclusions contains the outlook for application of the presented method to glacierized mountain regions worldwide. Figures and Tables good are balanced with the results in the main text.

**Thank you very much for taking the time to review our manuscript! The comments significantly helped to improve the presentation & discussion of results in this study as well as the general legibility.**

**RC.01. = Comments of referee 1**

*__General Comments__*

**RC.01.01:** As mentioned in L185, changes in glacier area and volume were very small in the 1970s and 1980s. This may have been more valid for smaller glaciers, such as those in the eastern Alps, than on large valley glaciers in the western part of the Alps. For the Swiss Alps, the volume change between 1970 and 2003 is about 20.9 km³, which corresponds to the values of 22.51 ± 1.76 km³ presented by e.g. Fischer et al. (2015). However, the volume change calculated from the total glacier volume presented for Austria (1970: 28.7±7.3km³; 2003: 13.3±3.8km³) is more than half of the original 1970 glacier volume. In comparison, Lambrecht et al. 2007 found a volume of 22.8 km3 for 1969 (GI 1) and 17.7 km3 for 1998 (GI 2), corresponding to a loss of 22%. Helfricht et al. (2019) presented a glacier volume loss of 6.4 km³ or 29% of the original volume from 1969 to 2006 (1969: 22.3 km³, 2006 15.9km3). The distinctly higher volume loss presented in this study results in part from the high volume in 1970. This should be discussed by the authors, especially based with respect to different glacier sizes.

> **Response: We agree with the reviewer that the differences in reported glacier volumes of the Austrian Alps are large and should be addressed in more detail. Therefore, we (1) revised the ice thickness reconstruction of the Austrian Alps, (2) computed the geodetic volume change of the studied period directly from the input DEMs (DHM69 - SRTM) and (3) analyzed differences regarding the glacier inventories used by this and previous studies.**

> **As mentioned above by the reviewer, the total glacier volumes for 1969 (22.3 km³), 1998 (19.7 km³) and 2006 (15.9 km³) by Helfricht et al. (2019) are distinctly smaller than the 1969 (~28 km³) and larger than the 2000 (~14 km³) volumes by this study, respectively.**

> > **(1) Regarding the very large difference between the 1969 Austrian glacier volume of this and previous studies (Lambrecht and Kuhn, 2007; Helfricht et al., 2019), we checked the calculation of the regional glacier volumes as presented in Table 1. Thereby, we noticed an error which caused a substantial overestimation of the 1969 volume. All ice thickness maps presented in this study were created in UTM projection. Since glaciers of the Austrian Alps are partially located in the UTM zones 32N and 33N, we created two ice thickness reconstructions for the respective UTM zone. Unfortunately, some of the glaciers which are located close to the UTM border between 32N and 33N were created by both reconstructions, resulting in an overestimation of the total glacier volume. This error in the Austrian Alps did not bias the derived viscosity scaling functions, these are based on the individual glacier viscosity fields and not on the total regional volumes, but caused a very large glacier volume in 1969 and therefore a bias in the volume comparison in section 4.3. The actual volume estimate is 23.9 km³ which is ~15% smaller than the original value but still higher than the volume reported by Lambrecht and Kuhn (2007) and Helfricht et al. (2019).**

> **We assume that the remaining differences in 1969 and 2000 glacier volumes are most likely related to the used elevation change data (2) and glacier outlines (3):**

**(2)** To evaluate the surface elevation change maps provided for the thickness reconstruction we refer to a previous study on glacier elevation and geodetic volume changes in the Austrian Alps by Lambrecht and Kuhn (2007). In this study, photogrammetric DEMs were used to calculate glacier volume changes of the Austrian Alps between 1969 and 1998. Thereby, the authors reported an overall change in glacier volume of -4.9 km³ during this period. Probably, this value can be seen as a lower bound of volume change during this period because for a small fraction of the 1969 Austrian glacier inventory, it was not possible to derive elevation changes due to insufficient quality of some of the aerial photographs (Lambrecht and Kuhn, 2007). Based on the DHM69 and SRTM DEM used in this study, we calculate a volume change rate of $-0.207\pm0.037$ km³a$^{-1}$ within the period 1969-2000, which is very similar to the observation period of Lambrecht and Kuhn (2007). Multiplied by ~29 years, the total volume change between 1969 and 1998 is approximately $-6.0\pm1.1$ km³ which is slightly more negative than the calculation by Lambrecht and Kuhn (2007). The remaining negative offset between the volume changes could be related to a negative elevation bias of the SRTM DEM at high altitudes which has been observed in the European Alps (Berthier et al., 2006; Paul, 2008) **and elsewhere** (Kumar et al., 2020). **Another explanation could be the slightly different observation periods or the calculation of elevation change from DEM acquisitions of different seasons (DHM69: autumn, SRTM: winter), which we cannot avoid in this comparison. However, this explanation is not very likely because the temporal offset between the observation periods is very small (< 2 years) and the annual glacier mass loss of the years 1999 & 2000 was not substantially higher than the mass change of previous years. Nevertheless, the error bars of our volume change estimates overlap with the reference value reported by Lambrecht and Kuhn (2007), indicating no substantial bias in the surface elevation change data used in this study for the Austrian Alps.**

**(3)** Further differences between the ice thickness reconstructions result from the used glacier inventories. Large differences in glacier area between the Austrian glacier inventories (GI 1-3) and Randolph Glacier Inventory (RGI) due to different outline delineations can be observed. The glacier volumes shown by Helfricht et al. (2019) for ~1998 (19.7 km³) and 2006 (15.9 km³) are based on glacierized areas of 470.7 km² (GI2) and 414.1 km² (GI3), respectively. The total glacier area of the Austrian Alps indicated by the RGI (~2003) is significantly smaller than both numbers. The RGI area sum of all glaciers which are at least partially located in Austria is 394.7 km². An example for these substantial differences between the available outlines is shown in Figure R1 and Table R1 below.

[Figure]

*Figure R1: Comparison of different outlines of glaciers of the Glocknergruppe, Austrian Alps: GI1 1969 (Patzelt, 1980), GI2 1998 (Lambrecht and Kuhn, 2007), RGI 2003 (Paul et al., 2011) & GI3 2009 (Fischer et al., 2015). Background: SRTM Hillshade.*

*Table R1: Number of glaciers and area of glacier samples shown in Figure R1: "Year" refers to the acquisition year of the respective glacier outlines as indicated in the provided metadata. The "Relative difference to GI2 area" column indicates the difference in area between the GI2 inventory and each other inventory (in percentages).*

| Inventory | Year | N. Glaciers | Area [km²] | Relative difference to GI2 area | Authors |
|---|---|---|---|---|---|
| GI1 | 1969 | 79 | 68.9 | + 16% | Patzelt (1980) |
| GI2 | 1998 | 78 | 59.8 | 0% | Lambrecht & Kuhn (2007) |
| RGI | 2003 | 66 | 46.9 | - 22% | Paul et al. (2011) |
| GI3 | 2009 | 78 | 51.7 | - 14% | Fischer et al. (2015) |

**Additionally, we initially reconstructed all Alpine glacier areas in this study and used thereafter a country border shapefile (Natural Earth 2022) to crop the glacierized areas to the respective country borders. This was necessary to provide a value for each country. For Austria, RGI glacier area of ~360 km² which is 23% and 13% less than the respective areas of the GI2 & GI3. Presumably, these differences are at least partially caused by a different delineation approach and differences in the interpretation of glacier extents, as stated by previous studies (Paul et al., 2011; Fischer et al., 2015), and ,potentially, the temporal difference and glacier retreat between the GI2 (1998) and RGI (2003). However, it is striking, that the RGI glacier**

area is even smaller than the more recent glacier inventory GI3 which might indicate that glacier areas of the Austrian Alps are in general slightly underestimated by the RGI.

All in all, we conclude that a rate of volume change between 1969 and ~2000 should not be directly derived from the numbers which we provided in Table 1. Likewise, it is difficult to directly compare early 21$^{st}$ century glacier volumes based on the Austrian GI2&3 (Lambrecht and Kuhn, 2007; Helfricht et al., 2019) and based on the RGI (this study). The reason for this is that the RGI glacier area, on which our second reconstruction step is based on, is much smaller than the glacier areas of the Austrian GI2 (1998) as well as GI3 (2006). This difference in glacier area is caused by technical differences between the inventories and the masking of the RGI glacier area to the exact country borders of Austria (3). The latter is necessary for our alpine-wide results as otherwise the sum of glacier volumes of the individual countries would be larger than the alpine-wide total volume.

To estimate the ~2000 glacier volume for the Austrian Alps of this study based on the Austrian inventory (GI2) instead of the RGI, we calculated the mean ice thickness of glacier areas which are included in the RGI and GI2 (~75% of the total GI2 area) and multiplied by the total glacier area of the GI2. With this approach, the approximated glacier volume for the Austrian Alps is 17.4 km³ which is more similar to the previously reported values of 17.7 km³ (Lambrecht and Kuhn, 2007) and 19.7 km³ (Helfricht et al., 2019) for the 1998 glacier area of the GI2.

In 1969, the difference in reconstructed ice thickness is more difficult to explain. The revised volume (1) of this study (~24 km³) is higher than the result of Helfricht et al. (2019), although both studies are based on the same glacier inventory (GI1). The higher volume in 1969 of this study might be caused by the observed slight overestimation of the glacier volume (and mass) loss since 1969 as shown above (2). Potentially, the reconstruction produces a higher 1969 glacier volume due to a more negative mass turnover as interfered from the surface elevation change data. Thereby, an overestimation of the surface elevation change rate since 1969 would also result in a higher glacier volume at the beginning of the observation period.

Changes in main manuscript: To address these volume differences in the manuscript, we (1) revised the total 1969 glacier volume of this study in section 4.1, 5 and Table 1 as well as the derived volume changes for the Austrian Alps. Additionally, we extended the volume comparison in section 4.1 by the following paragraphs to discuss the remaining differences in glacier volume and areas (2 & 3):

L433: *"Particularly for the Austrian Alps, large differences in regional glacier extents can be observed between the RGI glacier areas used in this study (Paul et al., 2011) and the Austrian glacier inventories GI2 and GI3 (Lambrecht and Kuhn, 2007; Fischer et al., 2015). Spatial differences in the delineation of glacier outlines appear to be mostly connected to small- and medium-sized glaciers and might be, at least partially, explained by the integration of perennial snowfields in the Austrian inventories as described by previous studies (Lambrecht and Kuhn, 2007; Paul et al., 2011). Therefore, a direct comparison of the GI2 and RGI glacier areas is difficult as reported by previous studies (Paul et al., 2011; Fischer et al., 2015). In addition, the RGI glacier area attributed to the Austrian Alps by this study is further reduced as we masked all glacierized areas to the Austrian country border in order to derive the*

*specific glacier volumes of each Alpine country (Table 1). We assume that the large differences in glacier volume for the early 21st century by this study and Helfricht et al. (2019) are related to the substantial differences in glacier area of the used inventories.*

*Assessing the difference in Austrian glacier volumes for 1969 by this and a previous study (Helfricht et al., 2019) is more complex since both studies are based on the same glacier inventory (GI1). Between 1969 and 1998, a volume change of -4.9 km³ was measured by Lambrecht and Kuhn (2007) based on DEM-differencing. For the very similar observation period 1969-2000 we derive a volume change rate of $-0.21\pm0.04$ km³a$^{-1}$ from the input DEMs (SRTM – DHM69) used in this study (section 2.4.1) which results in a more negative total volume change of $-6.0\pm1.1$ km³ for the reference period 1969-1998, which might be related to a negative elevation bias in the SRTM DEM at high altitudes (e.g. Berthier et al., 2006). Therefore, a potential explanation for the higher 1969 glacier volume in this study might be an overestimation of the surface elevation changes, and thus mass loss, of Austrian glaciers since 1969."*

**In Table 1, p.17 (presentation of glacier volume results) we added a remark with annotations (*) in the table that the volume numbers are based on different glacier inventories with large differences in glacier area due to methodological differences and should not be used directly to derive volume change rates.**

**RC.01.02:** Despite the regional differences in glacier volume, the local differences due to the different methods are obvious in figure 7. At Pasterze, for example, the highest estimate results from the presented method using retreat observations. However, the glacier appears to have a fairly thick tongue fed by a broad but relatively shallow upper cirque known from GPR measurements. The overestimation of ice thickness at higher elevations of the glacier causes a range of ice volume estimates by almost a factor of two, with the approach of this study at the upper end. Possibly, this is also true for other glaciers with similar topographic features (Rhone, Findel, ...). The same is shown in Figure 6 g vs. i, where bluish colors dominate the upper and lower parts of the Pasterze. But also, smaller glaciers southwest of the Pasterze show maximum ice thicknesses of up to 150 m in Figure 6g. A higher ice thickness and thus a higher ice volume also seems to be modelled for the Aletsch glacier region when comparing the presented approach (Fig. 6b) and the estimate constrained to measured ice thickness (Fig. 6d). It can be concluded that an overestimation of glacier volume may be calculated for regions where most of the ice thickness change data for calibration come from still thick glacier tongues of typical valley glaciers.

**Response: Local biases in estimated ice thickness due to the regionally-calibrated correction functions present a large uncertainty for individual glaciers as shown in Fig. 6 & 7 and discussed in section 4.3 for Aletsch glacier. We deliberately included the Pasterze in Figure 6 because the upper part of the glacier is one of the most prominent examples for a significant overestimation of the actual ice thickness by the reconstruction based on retreat thickness and viscosity correction (Fig. 6g). As mentioned above and in section 4.3, this bias is often found for flat glacier parts at high elevation and is in most cases introduced by the slope function. With the here presented regionally calibrated correction terms, it is very difficult to avoid these local biases if no direct thickness measurements are available (also compare Fig. 6f & g).**

Additionally, we extended the discussion of thickness biases due to the scaling functions in section 4.3 by illustrating the potential overestimation of ice thickness for the example of the upper parts of the Pasterze (p.24, l.516):

*"The initial slope-dependent viscosity correction ($\xi\eta slope$) shows a tendency of overestimating the ice thickness at high elevation. This pattern can be directly observed in the ice thickness maps (Fig.5 & 6). This is likely caused by the relatively large and flat accumulation areas of some Alpine glaciers where a high correction factor is applied to the ice flux. A very prominent example for such a significant overestimation of the actual ice thickness by $H_{SIA2003}^{retreat}$ can be observed at the Pasterze in Fig. 6f & g. For the large and mostly flat upper part of Pasterze, high viscosity values are estimated by the regionally calibrated slope-dependent correction. Without direct thickness measurements, it is very difficult to avoid these local biases."*

**RC.01.03:** The more general question is whether the scaling parameters and its thresholds, which are derived primarily from low-lying glacier tongues, are universally applicable to entire mountain regions with different assambling of glacier types. Please include this in more detail in your discussion.

**Response: This is an important suggestion and probably one of the main uncertainties which, unfortunately, we cannot completely avoid in this study. We suggest to extend the discussion in section 4.3 at l540 by:**

*"Eventually, the presented approach could be most beneficial in regions with large glacierized areas and sparse thickness observations where the glacier volume has to be interfered mostly from remote sensing information. However, another potential source of uncertainty, regarding the transferability of the presented correction terms to glacierized areas outside the European Alps, is related to the varying regional glacier morphologies in terms of size composition and elevation range. While the found empirical relations between ice viscosity and glacier surface topography have been applied to a different observation period and larger study region ($H_{SIA2003}^{retreat}$), we expect that the scaling functions are to some degree related to the geometries and size distribution of glaciers in the Swiss and Austrian Alps. In the European Alps, this uncertainty cannot be avoided because the overall distribution of a large number of small to medium-sized cirque glaciers with few large valley glaciers remains similar between $H_{SIA1970}^{retreat}$ and $H_{SIA2003}^{retreat}$, despite the substantial reduction in glacierized area since the 1970s. Further, the presented relations might be linked to the geographic environment of the European Alps as glacier changes are connected to the surrounding topography and climatic conditions (Abermann et al., 2011). To quantify this relation between the geometries of Alpine glaciers and the derived scaling parameters and to examine the transferability, it would be mandatory to extend the presented analysis to other glacierized regions with different glacier morphologies, such as marine-terminating glaciers, as well as different climatic settings, which is beyond the scope of this work."*

**RC.01.04:** L164 Please provide information on which resampling method was used.

> **Response: A bilinear resampling method was used. We changed the sentence accordingly:**
> ***"Finally, all elevation change fields were bilinearly resampled to a spatial resolution of 30m."***

**RC.01.05:** L170 Please present a value of the uncertainty per year relative to the mean annual ice thickness change (or relation of the total values for the 50y-period).

> **Response: We added respective relative uncertainty values.**

**RC.01.06:** L204 and L214 Please present in addition the relative value to the mean of the measured ice thickness

> **Response: We added the relative uncertainty of the in-situ and retreat thickness observations in l.204 & l.214.**

**RC.01.07:** L276 Eq. 5 With respect to the slope threshold, this equation may need a validity indication. For slopes smaller the threshold, the result may become negative else?!

> **Response: Yes, this is an important point. We added " … for α ≤ $\alpha_\eta^{thres}$ " to Eq.5 as suggested in the comment below.**

**RC.01.08:** L295 Eq.6 Same like in Eq. 5, a range of validity should be given (h≥h$_{tres}$?)

> **Response: In contrast to Eq.5, Eq.6 is valid and applied across the entire (normalized) elevation range between 0.0 and 1.0. We added therefore "… for 0 ≤ h ≤ 1.0" to Eq. 6.**

**RC.01.09:** L273 to L293 Please consider to move the two equations to Chapt. 2.2

> **Response: We agree that it may be favorable to have all equations in the same section. However, with the current structure, the presentation of Eq. 5 & 6 in sections 2.5.1 is closely followed by Figure 4 & 5 in section 3.1 which show the calculation of those equations. Thereby, section 2.2 provides a rather general introduction to the underlying approach (from previous studies). After the presentation of the input data, section 2.5 gives a detailed overview of the experimental setup and the approach developed in this study, followed by the presentation of the specific outcomes. Therefore, we would prefer to keep both equations in section 2.5.1 to preserve this separation between the general reconstruction method (of previous studies) and the new developments of this study in section 2.5 and 3.**

**RC.01.10:** Figure 2: Please increase the font used in the legend.

> **Response: Legend fonts were increased.**

**RC.01.11:** Figure 6: Partly it is hard to see the main differences apart the big blue Aletsch glacier. One option would be to present difference maps between b/g and the other calculations in c/d/e and h/i/j using a different colour grading differentiating negative and positive differences.

> **Response: Agree, difference maps between the retreat thickness setup of this study and the reference studies are another interesting illustration to show the local deviations in estimated ice thickness. As example we have created a respective figure for Aletsch glacier below (same figure for Pasterze will be added during revision).**
>
> **We replaced the lower panels of Fig. 6 with the difference maps of Aletsch (example below) Fig. 6 b-e) and Pasterze Fig. 6 g-j). The original panels of the ice thickness maps (as provided by the reference studies) were moved to a new figure in the supplement.**

[Figure]

*Figure 2: Reconstructed ice thickness ($H^{retreat}_{SIA2003}$) of Aletsch (b) and Pasterze glacier (g). Vertical differences in ice thickness between $H^{retreat}_{SIA2003}$ and previous studies for Aletsch glacier are shown in panel (c) Farinotti et al. 2019a, (d) Grab et al. 2021 and (e) Millan et al. 2022. (Background: SRTM hillshade)*

**RC.01.12:** Figure 7: Please add the axis title for ice thickness.

> **Response: We added "Mean ice thickness [m]" to the x-axis**

**References - responses**

Abermann, J., Kuhn, M., and Fischer, A.: Climatic controls of glacier distribution and glacier changes in Austria, Ann. Glaciol., 52, 83–90, https://doi.org/10.3189/172756411799096222, 2011.

Berthier, E., Arnaud, Y., Vincent, C., and Rémy, F.: Biases of SRTM in high-mountain areas: Implications for the monitoring of glacier volume changes, Geophys. Res. Lett., 33, L08502, https://doi.org/10.1029/2006GL025862, 2006.

Fischer, A., Seiser, B., Stocker Waldhuber, M., Mitterer, C., and Abermann, J.: Tracing glacier changes in Austria from the Little Ice Age to the present using a lidar-based high-resolution glacier inventory in Austria, The Cryosphere, 9, 753–766, https://doi.org/10.5194/tc-9-753-2015, 2015.

Helfricht, K., Huss, M., Fischer, A., and Otto, J.-C.: Calibrated Ice Thickness Estimate for All Glaciers in Austria, Front. Earth Sci., 7, 68, https://doi.org/10.3389/feart.2019.00068, 2019.

Kumar, A., Negi, H. S., Kumar, K., and Shekhar, C.: Accuracy validation and bias assessment for various multi-sensor open-source DEMs in part of the Karakoram region, Remote Sens. Lett., 11, 893–902, https://doi.org/10.1080/2150704X.2020.1792001, 2020.

Lambrecht, A. and Kuhn, M.: Glacier changes in the Austrian Alps during the last three decades, derived from the new Austrian glacier inventory, Ann. Glaciol., 46, 177–184, https://doi.org/10.3189/172756407782871341, 2007.

Patzelt, G.: The Austrian glacier inventory: status and first results, IAHS-AISH Publ., 1980.

Paul, F.: Calculation of glacier elevation changes with SRTM: is there an elevation-dependent bias?, J. Glaciol., 54, 945–946, https://doi.org/10.3189/002214308787779960, 2008.

Paul, F., Frey, H., and Le Bris, R.: A new glacier inventory for the European Alps from Landsat TM scenes of 2003: challenges and results, Ann. Glaciol., 52, 144–152, https://doi.org/10.3189/172756411799096295, 2011.